# Detecting Influence Structures in Multi-Agent Reinforcement Learning

## Abstract

We consider the problem of quantifying the amount of influence one agent can exert on another in the setting of multi-agent reinforcement learning (MARL). As a step towards a unified approach to express agents' interdependencies, we introduce the total and state influence measurement functions. Both of these are valid for all common MARL systems, such as the discounted reward setting. Additionally, we propose novel quantities, called the total impact measurement (TIM) and state impact measurement (SIM), that characterize one agent's influence on another by the maximum impact it can have on the other agents' expected returns and represent instances of impact measurement functions in the average reward setting. Furthermore, we provide approximation algorithms for TIM and SIM with simultaneously learning approximations of agents' expected returns, error bounds, stability analyses under changes of the policies, and convergence guarantees. The approximation algorithm relies only on observing other agents' actions and is, other than that, fully decentralized. Through empirical studies, we validate our approach's effectiveness in identifying intricate influence structures in complex interactions. Our work appears to be the first study of determining influence structures in the multi-agent average reward setting with convergence guarantees.

## 1 Introduction

The knowledge of mutual influence among a general system consisting of several entities, subsequently called agents, is beneficial to learn good strategies. The present work is regarding the influence among agents in the area of multi-agent reinforcement learning (MARL). Here, a shared environment is affected by the joint action of multiple agents. For each state of the environment, each agent chooses an action from its action space. The resulting joint action determines the transition to the following state. Each agent receives a reward for each transition, which is allowed to be different for every agent. Here, we consider the problem of giving a unified representation and an interpretable and measurable quantification of influence among agents.

Existing work addresses specific use cases and objectives of influence structures in MARL systems, such as reducing the number of agents that need to collaborate (Guestrin et al., 2002a), guiding exploration to states with high influence (Wang et al., 2020), or determining which agents need to communicate (Jaques et al., 2018). They focus on analyzing their method's effect on the system's objective without explicitly addressing the influence measurement's common theoretical aspects. Furthermore, the mentioned methods to measure influence among agents are exclusively focusing on the discounted reward setting (Sutton & Barto, 2018). As such, there is a lack of research related to influence in the average reward setting (Puterman, 1994), which is typically used for ongoing applications, e.g., load management in energy networks (Callaway & Hiskens, 2011), formation control of vehicles (Fax & Murray, 2004), or repeated auctions (Hoen et al., 2005).

The existing approaches mentioned above seek to resolve specific problems, such as a reduction of the joint action space by using a proxy of agents' influence on one another. While our method can be used for these applications as well, the main goal of our work is to address the fundamental question of how to reliably detect the inherent influence structure of an environment given a specific policy.

The main contributions of our work are the following. We introduce a unified approach to express a multi-agent system's inherent influence structure, regardless of the reward setting and overall objective. We then build upon this foundation by introducing the *total impact measurement* and *state*

*impact measurement*. These measurements quantify the overall and state-dependent influence structure, respectively, in the multi-agent average reward setting. In addition, we provide decentralized algorithms with stability analysis and convergence guarantees along with complementary empirical evaluations. To the best of our knowledge, our work is the first study of determining influence structures in the multi-agent average reward setting with provable convergence guarantees.

## 2 RELATED WORK

One popular representation of agents' dependencies is a coordination graph (Guestrin et al., 2002a), which is used to determine which agents' actions are relevant for the individual state-action functions. Several works try to detect the influence that the agents can exert on one another, e.g., (Kok et al., 2005). In contrast to our method, they require storing all estimations of the state-action values for the whole time horizon. Furthermore, they do not provide any theoretical analysis of their approximation method's quality. Another approach estimates the maximum expected utility one agent can potentially receive when coordinating with subgroups (Zhang & Lesser, 2013). Unlike our method, they rely on an approximation of the state transition probabilities of the underlying Markov decision process and only provide empirical evaluations for their method.

Wang et al. (2020) introduce the *Value of Interaction* to guide exploration to relevant states. Their formulation is similar to our proposed formulation of dependencies among agents. However, they rely on empirical estimation of the state transition probabilities, which is not the case for our work. Furthermore, their formulation is restricted to a specific state, whereas TIM, as proposed in this work, is formulated for the overall influence of one agent on another. More recently, researchers use the variance in state-action functions to construct context-aware coordination graphs (Wang et al., 2021). Contrary to our work, they do not provide any error bounds of their approximation quality and their formulation is again restricted to specific states only.

Instead of examining the influence between agents via their ability to alter the expected long-term return, Jaques et al. (2018) define causal influence by the changes of one agent's actions in the policy of another. However, their approach either demands that the probability of another agent's action, given a counterfactual action, is known or estimated. Our approach does not rely on this information, as we only require observing the other agents' actions.

## 3 BACKGROUND

This section introduces the multi-agent Markov decision process (MDP) in the infinite horizon average reward setting. It is the natural extension of the single-agent case introduced by Puterman (1994), and is also known as Markov game (Littman, 1994). In the second part, we present some results from stochastic approximation (Borkar, 2008), which we need to prove our main results.

### 3.1 MULTI-AGENT MDP

We consider a system of $N$ agents operating in a shared environment with discrete time steps $t \in \mathbb{N}$. The set of agents is denoted by $\mathcal{N}$. The environment can be described by a multi-agent MDP, which we specify in the following definition.

**Definition 3.1.** A multi-agent MDP is defined by a tuple $(\mathcal{S}, \{\mathcal{A}^i\}_{i \in \mathcal{N}}, P, \{r^i\}_{i \in \mathcal{N}})$, where $\mathcal{N} = \{1, \ldots, N\}$ denotes the set of agents, $\mathcal{S}$ is a finite state space which is shared by all agents, $\mathcal{A} = \prod_{i \in \mathcal{N}} \mathcal{A}^i$ is the joint action space, where $\mathcal{A}^i$ denotes the set of actions of agent $i$. Additionally, $P : \mathcal{S} \times \mathcal{A} \times \mathcal{S} \to [0, 1]$ is the MDP's state transition probability. There exist functions $R^i : \mathcal{S} \times \mathcal{A} \to \mathbb{R}$ with $R^i(s, a) = \mathbb{E}[r^i_{t+1} | s_t = s, a_t = a]$, which are denoted as the individual reward functions. Furthermore, the states and the joint actions are observable by every agent.

For every time step, each agent chooses its action according to its policy $\pi^i$, which is a probability distribution over $\mathcal{A}^i$. The joint policy is given by $\pi(s, a) = \prod_{i \in \mathcal{N}} \pi^i(s, a^i)$ for every $s \in \mathcal{S}$ and $a \in \mathcal{A}$. For a subset of agents $B^j = \{b^j_1, \ldots, b^j_k\} \subset \mathcal{N}$ we denote $a^{Bj} = (a^{b^j_1}, \ldots, a^{b^j_k})$, and $-B^j = \mathcal{N} \setminus B^j$. We are concerned with the average reward setting. The individual expected

time-average reward of agent $i \in \mathcal{N}$ is given by

$$J^i(\pi) := \lim_{T \to \infty} \frac{1}{T} \sum_{t=0}^{T-1} \mathbb{E}\left[r_{t+1}^i\right]. \tag{1}$$

To quantify the effects of a specific state and joint action, we define the relative individual state-action function for agent $i \in \mathcal{N}$, state $s \in \mathcal{S}$, and joint action $a \in \mathcal{A}$ as

$$Q_\pi^i(s,a) := \sum_{t \geq 0} \mathbb{E}\left[r_{t+1}^i - J^i(\pi)|s_0 = s, a_0 = a\right]. \tag{2}$$

Consider states $s, s' \in \mathcal{S}$. The probability of transitioning from state $s$ to $s'$ given a joint policy $\pi$ is given by $P_\pi(s'|s) = \sum_{a \in \mathcal{A}} \pi(s,a) \cdot P(s'|s,a)$. This induces a Markov chain over the states $\{s_t\}_{t \geq 0}$ with transition matrix $P_\pi \in \mathbb{R}^{|\mathcal{S}| \times |\mathcal{S}|}$. We make the following assumption on this Markov chain and the joint policy.

**Assumption 3.2.** The policies satisfy $\pi_i(s, a^i) > 0$ for every $i \in \mathcal{N}, s \in \mathcal{S}$ and $a^i \in \mathcal{A}^i$. Moreover, for every joint policy $\pi$ the induced Markov chain over the states $\{s_t\}_{t \geq 0}$ is ergodic, i.e., it is irreducible and aperiodic.

By Theorem 4.1 on page 119 in the book of Seneta (2006), there exists a unique stationary distribution for any ergodic Markov chain. We denote the stationary distribution of the Markov chain over the states by $d_\pi$. Given some states $s, s' \in \mathcal{S}$ and joint actions $a, a' \in \mathcal{A}$, the probability to transition from $(s,a)$ to $(s',a')$ can be expressed by $P_\pi^{\mathcal{A}}(s', a'|s, a) = P(s'|s,a) \cdot \pi(s', a')$. This induces a Markov chain over the states and actions $\{(s_t, a_t)\}_{t \geq 0}$ with transition matrix $P_\pi^{\mathcal{A}} \in \mathbb{R}^{|\mathcal{S}| \cdot |\mathcal{A}| \times |\mathcal{S}| \cdot |\mathcal{A}|}$. Note that this Markov chain is ergodic (Zhang et al., 2018) and its stationary distribution is given by $d_\pi^{\mathcal{A}}(s, a) = d_\pi(s) \cdot \pi(s, a)$, for every $s \in \mathcal{S}$ and $a \in \mathcal{A}$. The existence of a stationary distribution simplifies the study of the MDP immensely (Puterman, 1994). One property that we use throughout this paper is a simplified representation of averages of functions that depend on an ergodic Markov chain (Zhang et al., 2018). For example, one can represent the individual long-term return defined in Equation 1 by

$$J^j(\pi) = \sum_{s \in \mathcal{S}} d_\pi(s) \sum_{a \in \mathcal{A}} \pi(s,a) \cdot R^j(s,a). \tag{3}$$

## 3.2 STOCHASTIC ITERATION APPROXIMATION

Our main results use the following statements of the field of stochastic approximation iteration and motivate our algorithms' design. We state here a special case of Corollary 8 and Theorem 9 on pages 74-75 of Borkar (2008). These special cases have been formulated by Zhang et al. (2018). Consider an $n$-dimensional stochastic approximation iteration

$$x_{t+1} = x_t + \gamma_t \left[h\left(x_t, Y_t\right) + M_{t+1} + \beta_{t+1}\right], t \geq 0 \tag{4}$$

where $\gamma_t > 0$ and $\{Y_t\}_{t \geq 0}$ is a Markov chain on a finite set $A$.

**Assumption 3.3.** We make the following assumptions:

(a) $h : \mathbb{R}^n \times A \to \mathbb{R}^n$ is Lipschitz in its first argument;

(b) $\{Y_t\}_{t \geq 0}$ is an irreducible Markov chain with stationary distribution $\pi$;

(c) The stepsize sequence $\{\gamma_t\}_{t \geq 0}$ satisfies $\sum_{t \geq 0} \gamma_t = \infty$ and $\sum_{t \geq 0} \gamma_t^2 < \infty$;

(d) $\{M_t\}_{t \geq 0}$ is a martingale difference sequence, satisfying for some $K > 0$ and $t \geq 0$

$$\mathbb{E}\left(\|M_{t+1}\|^2 \mid x_\tau, M_\tau, Y_\tau; \tau \leq t\right) \leq K \cdot \left(1 + \|x_t\|^2\right); \tag{5}$$

(e) The sequence $\{\beta_t\}_{t \geq 0}$ is a bounded random sequence with $\beta_t \to 0$ almost surely as $t \to \infty$.

If Assumption 3.3 holds, then the asymptotic behavior of the iteration in Equation 4 is related to the behavior of the solution to the ordinary differential equation (ODE)

$$\dot{x} = \bar{h}(x) = \sum_{i \in A} \pi(i) h(x, i). \tag{6}$$

Suppose the ODE in Equation 6 has a unique globally asymptotically stable equilibrium $x^*$, then we have the following theorems connecting this solution to the original algorithm 4.

**Theorem 3.4.** Under Assumption 3.3, if $\sup_{t \geq 0} \|x_t\| < \infty$ a.s., we have $x_t \to x^*$

**Theorem 3.5.** Under Assumption 3.3, suppose that $\lim_{c \to \infty} \frac{\bar{h}(cx)}{c} = h_\infty(x)$ exists uniformly on compact sets for some $h_\infty \in C(\mathbb{R}^n)$. If the ODE $\dot{y} = h_\infty(y)$ has the origin as the unique globally asymptotically stable equilibrium, then $\sup_{t \geq 0} \|x_t\| < \infty$ almost surely.

## 4 INFLUENCE REPRESENTATIONS

The present work aims to specify and detect influence structures among agents in a multi-agent system. For this purpose, we first specify dependent and independent agents, following the definition of Guestrin et al. (2002b). Afterward, we introduce a novel representation framework of agents' influence structures, which is valid for all common reward settings and MDP formulations.

### 4.1 DEPENDENCIES AND INDEPENDENCIES IN MULTI-AGENT SYSTEMS

Given a state $s \in \mathcal{S}$, one agent's actions are relevant for another, if these directly influence the reward of the other agent, or affect the state for the other agent and, therefore, influence the reward indirectly. Both effects are captured in the individual state-action functions. Let $B^j \subset \mathcal{N}$ be a subset of agents and $j \in \mathcal{N}$, then agent $j$ is exclusively dependent on the agents in $B^j$ in state $s \in \mathcal{S}$ if

$$Q_\pi^j(s, a^{B^j}, a^{-B^j}) = Q_\pi^j(s, a^{B^j}) \text{ for all } a \in \mathcal{A}. \tag{7}$$

If this holds for all $s \in \mathcal{S}$, then agent $j$ acts completely independent in the MDP from agents in $B^{-j}$.

### 4.2 INFLUENCE MEASUREMENT FUNCTIONS

A binary representation of the dependency group $B^j$ is given by so-called coordination graphs (Guestrin et al., 2002b). However, strict independence as defined above often does not hold, which leads to large $B^j$'s or even $B^j = \mathcal{N}$. Several approaches demonstrated that one can approximate the individual state-action functions quite well by assuming some agents to be independent of each other (Sunehag et al., 2018; Böhmer et al., 2019; Zhang & Lesser, 2013). That means $Q_\pi^j(s, a) \approx Q_\pi^j(s, a^{\hat{B}^j})$ for $\hat{B}^j \subsetneq B^j$. That indicates that not every agent in $B^j$ has equal influence on agent $j$'s individual state-action function. Therefore, one needs a representation that allows a more fine-grained distinction of influence to express these differences.

There is no single quantity to express influence in a multi-agent system, as it depends on the specific use case. However, the study of different kinds of influence structures offers great value as a descriptive inherent property of multi-agent systems. Therefore, we propose a general framework to express influence structures in the form of abstract functions that are only bound by the independence criterion from Equation 7. We introduce an expression of state-dependent and global influence structures with the so-called state and total influence measurement functions.

**Definition 4.1** (State and total influence measurement functions). Let $\Omega$ be an arbitrary set, $\mathcal{N}$ a set of $N$ agents with joint policy $\pi = \prod_{j \in \mathcal{N}} \pi^j$, and individual state-action functions $Q_\pi^1, \ldots, Q_\pi^N$. Furthermore, let $\Psi^{\mathcal{S}}, \Psi : \mathcal{S} \times \Omega \to [0, \infty)^{N \times N}$ be matrix-valued functions. For any $s \in \mathcal{S}$ and $\omega \in \Omega$ an entry $\Psi_{i,j}^{\mathcal{S}}(s, \omega) > 0$ if and only if there exist actions $a^{-i} \in \mathcal{A}^{-i}, a^i, \hat{a}^i \in \mathcal{A}^i$ such that $Q_\pi^j(s, a^{-i}, a^i) \neq Q_\pi^j(s, a^{-i}, \hat{a}^i)$, then $\Psi^{\mathcal{S}}$ is called a state influence measurement function of the system of agents $\mathcal{N}$. Similarly, if for any $\omega \in \Omega$ an entry $\Psi_{i,j}(\omega) > 0$ if and only if there exist a state $s \in \mathcal{S}$ and actions $a^{-i} \in \mathcal{A}^{-i}, a^i, \hat{a}^i \in \mathcal{A}^i$ such that $Q_\pi^j(s, a^{-i}, a^i) \neq Q_\pi^j(s, a^{-i}, \hat{a}^i)$. Then, the function $\Psi$ is called a total influence measurement function of the system of agents $\mathcal{N}$.

Note that the definitions of state and total influence measurement functions are valid for any setting with a well-defined individual state-action function. Therefore, it holds for the average reward setting, which we focus on in our later analyses, but also for the discounted reward setting (Sutton & Barto, 2018). Furthermore, it holds for setups with infinite state and action spaces. The set $\Omega$ offers a parametrization of an influence measurement, for example, in the form of a prior that holds expert knowledge about the environment.

The value of an influence measurement function's knowledge is contingent on its semantic meaning. Nonetheless, there are specific interpretations that are valid for any influence measurement function. For a total influence measurement function, one can assume that for every agent $j$ there exists at least one agent $i \in \mathcal{N}$ such that the individual state-action function $Q_\pi^j$ is dependent on the actions of agent $i$. Otherwise, no action in any state in the system could influence the reward of agent $j$ in any way. Note that $i = j$ is allowed here. That means that the matrix $\Psi(\omega)$ has a positive entry in any row and column. Therefore, one can always get either a row- or column-stochastic matrix $\overline{\Psi}(\omega)$ from $\Psi(\omega)$ by respectively normalizing the rows or columns.

For a column stochastic $\overline{\Psi}(\omega)$, one can interpret the column $j$ as a probability distribution of the influence the agents in $\mathcal{N}$ can have on agent $j$'s state-action function. In this case, one can deduce a ranking depending on $\Psi$, which means one can, e.g., determine which agents should be in the coordination group $B^j$. The entries in row $i$ in a row-stochastic matrix $\overline{\Psi}(\omega)$ can, on the other hand, be interpreted as a probability distribution of agent $i$'s influence on the system of agents according to $\Psi$. This can be used, for example, in a cooperative setting, where the objective is to maximize the long-term return of the whole system. An entry $\Psi_{i,j}(\omega)$ describes the influence agent $i$ has on agent $j$ according to $\Psi$. If this entry is large compared to the other ones in the row, then agent $i$ should pay attention to its effects on agent $j$'s expected reward when taking its actions.

The same deductions are valid for a state influence measurement function $\Psi^S$, although the assumption of a positive entry in every row and column does not necessarily hold.

## 5 INFLUENCE MEASUREMENT FUNCTIONS IN THE AVERAGE REWARD SETTING

We propose novel quantities to measure influence among agents, as the maximum impact an agent can have on the individual state-action function of another. We show that the proposed quantities are instances of a state and total influence measurement function respectively, and give approximation algorithms with convergence guarantees. We refer to Appendix A for the proofs.

### 5.1 THE TOTAL IMPACT MEASUREMENT

The core of the proposed measurements consists of the *impact* sample, which quantifies the maximum impact one agent can have on the return of another given a specific state and joint action.

**Definition 5.1** (Impact sample). Let $\pi = \prod_{i \in \mathcal{N}} \pi^i$ be a joint policy of a set of agents $\mathcal{N}$, which are acting in a multi-agent MDP, and denote with $Q_\pi^j$ the individual state-action function for agent $j$. For a state $s \in \mathcal{S}$ and joint action $a \in \mathcal{A}$, we define the impact sample of agent $i$ on agent $j$ as

$$U_\pi^{i \to j}(s,a) := \max_{a^i \in \mathcal{A}^i} Q_\pi^j(s, a^{-i}, a^i) - \min_{a^i \in \mathcal{A}^i} Q_\pi^j(s, a^{-i}, a^i). \tag{8}$$

The impact sample for agent $j$ on agent $i$ given a specific $s \in \mathcal{S}$ and a joint action $a \in \mathcal{A}$ indicates how much agent $j$ can influence the expected long-term return of agent $i$. Averaging this over all possible states and joint actions yields the total impact measurement.

**Definition 5.2** (Total impact measurement). Let $\pi = \prod_{i \in \mathcal{N}} \pi^i$ be a joint policy of the agents and $\{(s_t, a_t)\}_{t \geq 0}$ the induced Markov chain over the states and actions in a multi-agent MDP. The total impact measurement (TIM) of agent $i$ on agent $j$, for $i, j \in \mathcal{N}$, is then defined as

$$TI^{i \to j}(\pi) := \lim_{T \to \infty} \frac{1}{T} \sum_{t=0}^{T-1} \mathbb{E}\left[U_\pi^{i \to j}(s_t, a_t)\right]. \tag{9}$$

Note that under Assumption 3.2, there exists a stationary distribution over the states and actions $d_\pi^{\mathcal{A}} = d_\pi \cdot \pi$, where $d_\pi$ is the stationary distribution over the states. Then one can represent TIM by

$$TI^{i \to j}(\pi) = \sum_{s \in \mathcal{S}} d_\pi(s) \sum_{a \in \mathcal{A}} \pi(s, a) \cdot U_\pi^{i \to j}(s, a). \tag{10}$$

As the stationary distribution $d_\pi^{\mathcal{A}}$ is strictly positive and the impact samples $U_\pi^{i \to j}$ are greater or equal to zero, we see that $TI^{i \to j}(\pi) = 0$ if and only if $U_\pi^{i \to j}(s, a) = 0$ for all $s \in \mathcal{S}, a \in \mathcal{A}$. When we observe Equation 8, we see that $TI^{i \to j}(\pi) = 0$ if and only if $Q^j(s, a^{-i}, a^i) = Q^j(s, a^{-i}, \hat{a}^i)$ for all $s \in \mathcal{S}, a^{-i} \in \mathcal{A}^{-i}$ and $a^i, \hat{a}^i \in \mathcal{A}^i$. Therefore, the constant matrix-valued function $TI_\pi : \Omega \to [0, \infty)^{N \times N}$, with entries given by $(TI_\pi)_{i,j}(\omega) = TI^{i \to j}(\pi)$, is a total influence measurement function by Definition 4.1.

That means, if we can estimate TIM reliably, we obtain an unbiased total influence measurement function. Its semantic meaning is determined by the impact sample, i.e., it represents the maximum impact of an agent on the expected long-term return of another. In general, one does not know the individual state-action functions, but only some approximations of them. We denote an approximation of an individual state-action function by $\overline{Q}_\pi^j$ and a resulting formulation of an approximated TIM using Equation 10 by $\overline{TI}_\pi^{i \to j}$. The following theorem gives an error bound between the approximated TIM and the true TIM, depending on the individual state-action functions' approximation error.

**Theorem 5.3.** The error of the approximated TIM to the true one of agent $i$ on agent $j$ satisfies

$$\left| TI^{i \to j}(\pi) - \overline{TI}^{i \to j}(\pi) \right| \le 2 \cdot \left\| Q_\pi^j - \overline{Q}_\pi^j \right\|_\infty. \tag{11}$$

This bound shows that if we can determine $\overline{TI}^{i \to j}(\pi)$, we get a good approximation of TIM provided that the approximation error of $\overline{Q}_\pi^j$ is small. For an approximation function, we consider parametrized function classes. Denote with $Q_\pi^j : \mathcal{S} \times \mathcal{A} \times \mathbb{R}^{k_j} \to \mathbb{R}$ the individual state-action function of agent $j$, parametrized by $\eta^j \in \mathbb{R}^{k_j}$ for $k_j \in \mathbb{N}$. We denote the parametrized impact samples and TIM by $U_\pi^{i \to j}(s, a, \eta^j)$ and $TI^{i \to j}(\pi, \eta^j)$ respectively.

Our proposed approximation algorithm of TIM works together with a simultaneously learning state-action function approximation algorithm, which provides an iteration sequence $\{\eta_t^j\}_{t \ge 0}$. For our later results, we state two mild assumptions on the iteration algorithm creating $\{\eta_t^j\}_{t \ge 0}$ and the parametrized individual state-action functions.

**Assumption 5.4.** The parametrized state-action function $Q^j(s, a, \eta)$ is continuous in $\eta \in \mathbb{R}^{k_j}$, for every $j \in \mathcal{N}, s \in \mathcal{S}$, and $a \in \mathcal{A}$.

**Assumption 5.5.** Let $j \in \mathcal{N}$. We assume that the iteration sequence $\{\eta_t^j\}_{t \ge 0}$ is almost surely bounded, i.e., there exists a $K > 0$ such that $\sup_{t \ge 0} \left\| \eta_t^j \right\| < K < \infty$ almost surely. Additionally, there exists an $\eta^{j,*} \in \mathbb{R}^{k_j}$ such that $\eta_t^j \to \eta^{j,*}$ almost surely.

The above assumption essentially demands that the iteration algorithm, to approximate the individual state-action function, converges at some point. The considered iteration algorithm of TIM with parametrized individual state-action functions is given by

$$\nu_{t+1}^{i \to j} = (1 - \alpha_t)\nu_t^{i \to j} + \alpha_t \cdot U_\pi^{i \to j}(s_t, a_t, \eta_t^j), \tag{12}$$

where $\{\alpha_t\}_{t \ge 0}$ is a stepsize sequence satisfying part (c) of Assumption 3.3. With this, we can now state our main result.

**Theorem 5.6.** Under Assumptions 3.2, 5.4, and 5.5, the iteration defined in Equation 12 has the following convergence property

$$\nu_{t+1}^{i \to j} \to TI^{i \to j}(\pi, \eta_\pi^{j,*}) \text{ almost surely.} \tag{13}$$

## 5.2 THE STATE IMPACT MEASUREMENT

TIM averages the maximum impact one agent can have on the individual state-action function of another over all possible transitions. However, given a specific state, some agents might have a

significant impact on the individual state-action functions of others, even though their average influence is small. Therefore, one would like to quantify state-dependent influence structures among the agents. Therefore, we introduce the state impact measurement, which constitutes a state influence measurement function by Definition 4.1.

**Definition 5.7** (State impact measurement). Let $\pi$ be a joint policy of the $N$ agents over the joint action space $\mathcal{A}$. Take the state $s \in \mathcal{S}$ and denote the Markov chain over the actions in state $s$ by $\{a_{t^s}^s\}_{t^s \geq 0}$. The state impact measurement (SIM) of agent $i$ on agent $j$, for $i, j \in \mathcal{N}$ is defined as

$$SI^{i \to j}(s, \pi) := \lim_{T^s \to \infty} \frac{1}{T^s} \sum_{t^s=0}^{T^s-1} \mathbb{E}\left[ U_\pi^{i \to j}(s, a_{t^s}^s) \right]. \tag{14}$$

Note that SIM only considers the Markov chain over the actions $\{a_{t^s}^s\}_{t^s \geq 0}$ given a specific state $s \in \mathcal{S}$. Hence, one ignores the MDP's state transition probabilities and only considers the distribution over the joint actions for a state $s$. As we only consider the actions for a given state $s$, $\pi(s, \cdot)$ is the stationary distribution of the Markov chain $\{a_{t^s}^s\}_{t^s \geq 0}$. Therefore, one can represent SIM by

$$SI^{i \to j}(s, \pi) = \sum_{a \in \mathcal{A}} \pi(s, a) \cdot U_\pi^{i \to j}(s, a). \tag{15}$$

Under Assumption 3.2, one can record the instances of $\{a_t\}_{t \geq 0}$ for each state $s$ in a tabular fashion, which allows sampling from $\{a_{t^s}^s\}_{t^s \geq 0}$. With this insight, one can observe that the theoretical results from Subsection 5.1 carry over with only slight variations in the proofs. Therefore, we only state the following results without proof and refer to the supplementary material for more details.

First, we give an error bound similar to the statement from Theorem 5.3. We denote the approximated SIM by $\overline{SI}^{i \to j}$ using the approximated individual state-action function $\overline{Q}_\pi^j$.

**Theorem 5.8.** Let $s$ be in $\mathcal{S}$. The error of the approximated SIM in $s$ to the true one of agent $i$ on agent $j$ satisfies

$$\left| SI^{i \to j}(s, \pi) - \overline{SI}^{i \to j}(s, \pi) \right| \leq 2 \cdot \left\| Q_\pi^j(s, \cdot) - \overline{Q}_\pi^j(s, \cdot) \right\|_\infty. \tag{16}$$

We denote the parametrized SIM by $SI^{i \to j}(\cdot, \pi, \eta^j)$ for $i, j \in \mathcal{N}$ and $\eta^j \in \mathbb{R}^{k_j}$. The tabular approximation algorithm is

$$\nu_{t^s+1}^{i \to j}(s) = (1 - \alpha_{t^s})\nu_{t^s}^{i \to j}(s) + \alpha_{t^s} \cdot U_\pi^{i \to j}(s, a_{t^s}^s, \eta_t^j), \tag{17}$$

where $\{\alpha_{t^s}\}_{t^s \geq 0}$ denotes a stepsize sequence satisfying part (c) of Assumption 3.3.

**Theorem 5.9.** Under Assumptions 3.2, 5.4, and 5.5, the iteration defined in Equation 17 has the following convergence property for every $s \in \mathcal{S}$

$$\nu_{t^s+1}^{i \to j}(s) \to SI^{i \to j}(s, \pi, \eta_\pi^{j,*}) \text{ almost surely.} \tag{18}$$

## 5.3 Continuity in Policy Changes

The preceding analyses treated the joint policy $\pi$ as fixed. In the following, we relax this restriction and show that TIM and SIM are continuous in changes of the joint policy $\pi$, which is crucial for practical applications as one can expect the approximation algorithm's behavior to be highly unstable otherwise.

We consider parameterized functions to track changes in the policies. Let $\theta^j \in \mathbb{R}^{m_j}$ for $m_j \in \mathbb{N}$ and $\pi_{\theta^j}^j$ be the policy of agent $j$. Denote with $\theta = [(\theta^1)^T, \ldots, (\theta^N)^T]^T \in \mathbb{R}^m := \prod_{j \in \mathcal{N}} \mathbb{R}^{m_j}$ the joint policy parameters, and denote the parametrized joint policy by $\pi_\theta = \prod_{j \in \mathcal{N}} \pi_{\theta^j}^j$. Note that when we require Assumption 3.2 to hold, that it is assumed that the parametrized policies have a positive probability for every state and action. Furthermore, we assume the following:

**Assumption 5.10.** The function $\pi_{\theta^j}^j(s, a^j)$ is continuously differentiable in $\theta^j \in \mathbb{R}^{m_j}$.

To prove the continuity of TIM and SIM in $\theta$, one needs to establish the continuity of the stationary distribution $d_\theta$, the joint policy $\pi_\theta$, and the impact samples $U_\theta^{i \to j}$.

**Theorem 5.11.** Let $\Theta \subset \mathbb{R}^m$ be a compact set, and let $\pi_\theta$ be the joint policy. Under Assumptions 3.2 and 5.10, $TI^{i \to j}(\pi_\theta)$ and $SI^{i \to j}(s, \pi_\theta)$ are continuous in $\theta \in \Theta$ for every $i, j \in \mathcal{N}$ and $s \in \mathcal{S}$.

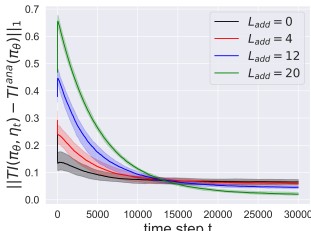 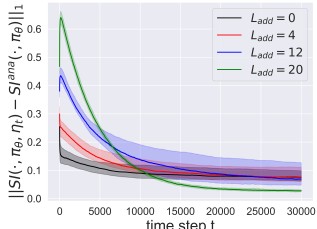 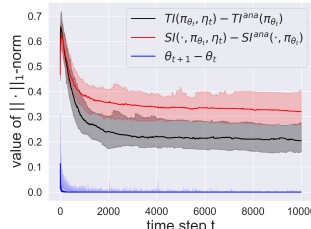

(a) TIM error with static $\pi_\theta$ and varying dependency structures

(b) SIM error with static $\pi_\theta$ and varying dependency structures

(c) TIM and SIM approximation error with learning $\pi_{\theta_t}$

Figure 1: Performance of TIM and SIM's approximation algorithms in the random environment, conducted over 50 seeds. The bold line represents the median, while the shaded areas denote the 95%-quantiles.

## 6 EMPIRICAL RESULTS

The stochastic approximation techniques applied to ensure the convergence of our proposed algorithms do not guarantee specific convergence behaviors in practice (Borkar, 2008). Thus, to better understand these behaviors, we evaluate our concepts in two contrasting environments. The first environment is a small, randomly generated one. The second environment is a multi-agent extension of the coin game (Lerer & Peysakhovich, 2017).

For both environments, we use iteration algorithms from Equations 12 and 17 to estimate TIM and SIM, initializing them for all $i, j \in \mathcal{N}$ to $\frac{1}{|\mathcal{N}|}$. Their approximations are represented by $TI(\pi_\theta, \eta_t)$ and $SI(s, \pi_\theta, \eta_t)$. All experiments employ Boltzmann policies (Sutton & Barto, 2018), meeting the Assumptions 3.2 and 5.10. Further details and supplementary results are available in the appendix.

### 6.1 RANDOM ENVIRONMENT

We generate a random multi-agent MDP with five agents, five states, and binary action spaces (see Section C.1 for details). We aim to understand how TIM and SIM approximation algorithms respond to different agent influence structures. Therefore, when we conduct experiments where some agents are independent of others, we set the rows of the transition probability matrix all equal to the first row, i.e., $P(s'|\cdot, \cdot) = P(s^0|\cdot, \cdot)$ for all $s' \in \mathcal{S}$. This prevents the agents to influence one another over long-term effects on the transitions to other states. Furthermore, to achieve that agent $j$ is independent of the immediate effects on the reward of agent $i$'s actions, we set the entries for a state $s \in \mathcal{S}$, and actions $a^{-i} = (a^1, \ldots, a^{i-1}, a^{i+1}, \ldots a^N) \in \mathcal{A}^{-i}$ in the reward matrix to $R^j(s, a^{-i}, a^i) = R^j(s, a^{-i}, \hat{a}^i)$ for all $a^i, \hat{a}^i \in \mathcal{A}^i$.

Experiments in the random environment took two forms. First, we consider a static policy $\pi_\theta$ for different dependency structures among the agents. We assume that each agent can at least influence its individual state-action function. To determine the overall dependency structures among the agents, we randomly sample a number of additional dependencies $L_{\text{add}}$. The second experiment has no enforced influence structure but changing policy parameters $\theta_t$. As the policies' learning algorithm, we use Algorithm 1 of Zhang et al. (2018), which is a multi-agent actor-critic algorithm for a fully cooperative setup. For approximating individual state-action functions $Q_\theta^j(\cdot, \cdot, \eta^j)$, we use the tabular SARSA algorithm in the average-reward setting (Sutton & Barto, 2018). Note that this algorithm satisfies Assumptions 5.4 and 5.5. We compare the approximations with analytically determined TIM and SIM matrices, which are denoted by $TI^{\text{ana}}(\pi_\theta)$ and $SI^{\text{ana}}(s, \pi_\theta)$.

The results with a stationary policy can be seen in Figures 1a and 1b. They show the approximation errors of $TI(\pi_\theta, \eta_t)$ to $TI^{\text{ana}}(\pi_\theta)$ and $SI(s, \pi_\theta, \eta_t)$ to $SI^{\text{ana}}(s, \pi_\theta)$ for different values of $L_{\text{add}}$. In all scenarios, the error is monotonically decreasing in $t$. One observes that the initial approximation error increases with an increasing number of dependencies among the agents. However, the final approximation error has the reversed order. This results from the fact that impact samples need to be zero to detect that two agents are independent. However, a non-zero approximation error in the individual state-action functions leads to an overestimation of the TIM and SIM approximations.

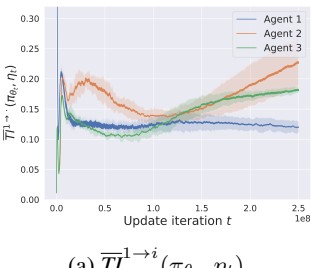 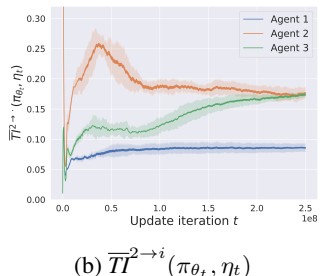 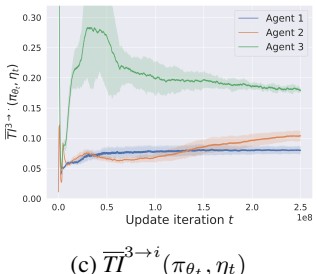

(a) $\overline{TI}^{1 \to i}(\pi_{\theta_t}, \eta_t)$      (b) $\overline{TI}^{2 \to i}(\pi_{\theta_t}, \eta_t)$      (c) $\overline{TI}^{3 \to i}(\pi_{\theta_t}, \eta_t)$

Figure 2: Mean TIM approximations in coin game over 10 seeds. The shaded area shows the standard deviation.

For the dynamic policy, results are in Figure 1c. It shows consistent, albeit slower, reduction in approximation errors compared to the static policy. Nonetheless, this experiment demonstrates the validity of using the approximation algorithms in the context of changing policies.

## 6.2 COIN GAME

Three agents navigate a $10 \times 10$ grid, with the actions being to move in one of four directions or to remain static. Unique coins, designated for each agent, randomly spawn on the grid, with up to four coins per agent at any given time. Collecting a coin grants an agent a reward of $1$. However, if an agent collects another's coin, penalties ensue. Deviating from the original game to emphasize asymmetries in agent-dependencies, we employ a one-sided penalty system. In our setting, if agent $1$ collects coins of agents $2$ or $3$, they incur a penalty of $-2$. Additionally, agent $3$ suffers a $-2$ penalty if agent $2$ collects its coins. Notably, agent $1$ remains unpenalized throughout, while agent $2$ only faces penalties from agent $1$'s actions.

For the experiment, agents independently learn using the PPO algorithm (Schulman et al., 2017). We utilize the deep SARSA algorithm (Zhao et al., 2016) to approximate $Q_\theta^j(\cdot, \cdot, \eta^j)$, which satisfies Assumption 5.4 but not necessarily Assumption 5.5. As the state space is intractable, we train an additional neural network to approximate SIM instead of using a tabular method.

The PPO policies learned to greedily gather coins, irrespective of coin type. The TIM approximations of agent $1$'s influence on other agents are depicted in Figure 2a. As anticipated, the estimated impact is high across all agents, especially considering agent $1$'s capacity to impose penalties on agents $2$ and $3$. Conversely, Figure 2b highlights agent $2$'s significant impact on itself and agent $3$, but minimal effect on Agent $1$, mirroring the unique penalty structure. Figure 2c underscores that TIM for Agent $3$ is predominantly elevated only for itself. Note that the absence of direct penalties does not render the agents independent. Agent $3$, for instance, possesses the capability to either block agent $1$ or seize its coins. Consequently, TIM estimations give us insights—without any knowledge of the environment—into which agent can significantly influence others.

## 7 FINAL REMARKS

The present work investigates influence structures in MARL systems. We introduce influence measurement functions as a unified descriptive framework for influence structures in all common setups. Within this framework, we propose total and state influence measures tailored to the average reward setting. Thorough theoretical analyses of their stability and the convergence and error bounds of the corresponding approximation algorithms are given. Experiments in the randomly generated environment demonstrate convergence of the approximation error, even with evolving policies. The coin game further demonstrates the applicability of the concepts to complex, dynamic settings and provides insight into influence in black-box environments.

Future work offers promising directions. A key area involves expanding the application of TIM and SIM beyond their current descriptive roles, using them to enhance learning processes within MARL. Another avenue is to investigate the potential of influence measurement functions in other environments, such as those with discounted reward or infinite state and action spaces.

## REPRODUCIBILITY STATEMENT

To ensure the reproducibility of our results, we provide detailed additional information in the appendix. We present five novel theoretical claims, each substantiated with thorough proofs found in Appendix A. For clarity, each subsection in this appendix directly references its corresponding theorem from the main text.

Detailed descriptions of the utilized environments, our algorithmic setup and code base are available in Appendix C. For a clear understanding of our experimental setup, we outline our hyperparameter selection strategy and specific choices in Appendix D.

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

# A  PROOFS FOR TIM AND SIM STATEMENTS

## A.1  PROOF OF THEOREM 5.3 (TIM ERROR BOUND)

*Proof.* Let $i, j \in \mathcal{N}$, then we see

$$\left| TI^{i \to j}(\pi) - \overline{TI}^{i \to j}(\pi) \right| \leq \sum_{s \in \mathcal{S}} d_\pi(s) \sum_{a \in \mathcal{A}} \pi(s, a) \cdot \left( \left| \max_{a^i \in \mathcal{A}^i} Q_\pi^j(s, a^{-i}, a^i) - \max_{a^i \in \mathcal{A}^i} \overline{Q}_\pi^j(s, a^{-i}, a^i) \right| \right.$$

$$\left. + \left| \min_{a^i \in \mathcal{A}^i} Q_\pi^i(s, a^{-i}, a^i) - \min_{a^i \in \mathcal{A}^i} \overline{Q}_\pi^j(s, a^{-i}, a^i) \right| \right)$$

$$\leq \sum_{s \in \mathcal{S}} d_\pi(s) \sum_{a \in \mathcal{A}} \pi(s, a) \left( 2 \cdot \left\| Q_\pi^j - \overline{Q}_\pi^j \right\|_\infty \right) = 2 \cdot \left\| Q_\pi^j - \overline{Q}_\pi^j \right\|_\infty,$$

which gives us the statement. □

## A.2  PROOF OF THEOREM 5.6 (CONVERGENCE OF TIM APPROXIMATION)

*Proof.* We define

$$h(\nu_t^{i \to j}, (s_t, a_t)) := U_\pi^{i \to j}(s_t, a_t, \eta_\pi^{j,*}) - \nu_t^{i \to j},$$
$$M_{t+1} := 0,$$
$$\beta_{t+1} := U_\pi^{i \to j}(s_t, a_t, \eta_t^j) - U_\pi^{i \to j}(s_t, a_t, \eta_\pi^{j,*}),$$

where we can see that the iteration algorithm

$$\nu_{t+1}^{i \to j} = \nu_t^{i \to j} + \alpha_t \cdot \left( h(\nu_t^{i \to j}, (s_t, a_t)) + M_{t+1} + \beta_{t+1} \right) \tag{19}$$

is equal to the iteration algorithm defined in Equation 12. To show the convergence of this iteration algorithm, we consider in the first step a slightly different algorithm. For this, observe that by Assumption 5.5, the sequence $\{\eta_t^j\}_{t \geq 0}$ is almost surely bounded. That means there exists $K > 0$ such that $P(\sup_{t \geq 0} \|\eta_t^j\| < K) = 1$. Define the error term $\tilde{\beta}_t := \mathbb{I}_{\{\sup_{t \geq 0} \|\eta_t^j\| < K\}} \cdot \beta_t$, where $\mathbb{I}_A$ denotes the indicator function on a set $A$. Define the following iteration algorithm with the restricted error sequence $\tilde{\beta}_t$

$$\tilde{\nu}_{t+1}^{i \to j} := \tilde{\nu}_t^{i \to j} + \alpha_{\nu,t} \cdot \left( h(\tilde{\nu}_t^{i \to j}, (s_t, a_t)) + \tilde{\beta}_{t+1} \right). \tag{20}$$

First, we check that parts (a) to (e) from Assumption 3.3 hold. The function $h : \mathbb{R}^{k_j} \times \mathcal{S} \times \mathcal{A} \to \mathbb{R}$ is Lipschitz continuous in its first argument, as it is even linear in its first argument, i.e.

$$|h(\nu, (s, a)) - h(\nu', (s, a))| = |\nu - \nu'| \text{ for any } s \in \mathcal{S}, a \in \mathcal{A},$$

and part (a) holds. By Assumption 3.2, the Markov chain $\{(s_t, a_t)\}_{t \geq 0}$ is ergodic, which means that it satisfies part (b). Furthermore, the stepsizes $\{\alpha_t\}_{t \geq 0}$ satisfy part (c). As the sequence $\{M_t\}_{t \geq 0}$ is the zero sequence, it is trivially a martingale difference sequence with a conditionally bounded norm, and satisfies part (d). Finally, for part (e), it remains to show that $\{\tilde{\beta}_t\}_{t \geq 0}$ is a bounded random sequence that converges to zero almost surely. For this, note that $\eta_t^j$ is uniformly bounded on the set $\{\sup_{t \geq 0} \|\eta_t^j\| < K\}$. By Assumption 5.4, we get that the parametrized impact samples

$U_\pi^{i\to j}(s, a, \eta^j)$ are continuous in $\eta^j$. In particular, as it is a continuous function on the compact set $\{\eta^j \in \mathbb{R}^{k_j} : \|\eta^j\| < K\}$, we get that it is bounded. Therefore, together with the convergence of $\eta_t^j \to \eta_\pi^{j,*}$, we get that $\{\tilde{\beta}_t\}_{t\geq 0}$ is a bounded random sequence that converges to zero. Therefore, Assumption 3.3 is satisfied for the iteration algorithm from Equation 20. Next, consider the ODE given by

$$\dot{\nu}^{i\to j} = \sum_{s\in\mathcal{S}} d_\pi(s) \sum_{a\in\mathcal{A}} \pi(s,a) \cdot h\left(\nu^{i\to j}, (s,a)\right)$$
$$= -\nu^{i\to j} + \sum_{s\in\mathcal{S}} d_\pi(s) \sum_{a\in\mathcal{A}} \pi(s,a) \cdot U_\pi^{i\to j}(s, a, \eta^{j,*})$$

and define the right-hand side as $f(\nu^{i\to j})$. We can see that $\nu^{i\to j} = \sum_{s\in\mathcal{S}} d_\pi(s) \sum_{a\in\mathcal{A}} \pi(s,a) U_\pi^{i\to j}(s, a, \eta^{j,*})$ is an equilibrium solution to the ODE above, and as $f$ is Lipschitz continuous, we get by the theorem of Picard-Lindelöf, see page 89 in the book of Adkins & Davidson (2012), that this solution is unique. Define the function $f_c(\nu^{i\to j}) = c^{-1} \cdot f(c\nu^{i\to j})$. Then $\lim_{c\to\infty} f_c(\nu^{i\to j}) = -\nu^{i\to j} =: f_\infty(\nu^{i\to j})$ exists and the ODE $\dot{\nu}^{i\to j} = f_\infty(\nu^{i\to j})$ has the origin as unique asymptotically stable equilibrium. Therefore, we get by Theorem 3.5 that $\sup_{t\geq 0} \|\nu_t^{i\to j}\| < \infty$ almost surely. Then, we can use Theorem 3.4 to conclude that

$$\tilde{\nu}_t^{i\to j} \to TI^{i\to j}(\pi, \eta_\pi^{j,*}) \text{ a.s.} \tag{21}$$

To extend this result to the original iteration sequence, observe that $\left\{\sup_{t\geq 0} \|\eta_t^j\| \geq K\right\}$ is a null-set. □

### A.3 Proof of Theorem 5.8 (SIM Error Bound)

*Proof.* Let $i, j \in \mathcal{N}$, then we see

$$\left| SI^{i\to j}(s, \pi) - \overline{SI}^{i\to j}(s, \pi) \right| = \left| \sum_{a\in\mathcal{A}} \pi(s,a) \left( U_\pi^{i\to j}(s,a) - \overline{U}_\pi^{i\to j}(s,a) \right) \right|$$

$$\leq \sum_{a\in\mathcal{A}} \pi(s,a) \left( \left| \max_{a^i\in\mathcal{A}^i} Q_\pi^j(s, a^{-i}, a^i) - \max_{a^i\in\mathcal{A}^i} \overline{Q}_\pi^j(s, a^{-i}, a^i) \right| \right.$$

$$\left. + \left| \min_{a^i\in\mathcal{A}^i} Q_\pi^i(s, a^{-i}, a^i) - \min_{a^i\in\mathcal{A}^i} \overline{Q}_\pi^j(s, a^{-i}, a^i) \right| \right)$$

$$\leq \sum_{a\in\mathcal{A}} \pi(s,a) \left( 2 \cdot \left\| Q_\pi^j(s, \cdot) - \overline{Q}_\pi^j(s, \cdot) \right\|_\infty \right) = 2 \cdot \left\| Q_\pi^j(s, \cdot) - \overline{Q}_\pi^j(s, \cdot) \right\|_\infty,$$

which gives us the statement. □

### A.4 Proof of Theorem 5.9 (Convergence of SIM Approximation)

*Proof.* Let $s \in \mathcal{S}$. Note that for any subsequence $\{\eta_{t_k}\}_{k\geq 0}$ of $\{\eta_t\}_{t\geq 0}$ it still holds that $\eta_{t_k} \to \eta_\pi^{j,*}$ almost surely. Therefore, we denote the subsequences originally indexed by $t^s$ for state $s$ also by $t$ instead, as it does not change the proof's statement. We define

$$h(\nu_t^{i\to j}(s), a_t^s) := U_\pi^{i\to j}(s, a_t^s, \eta_\pi^{j,*}) - \nu_t^{i\to j}(s),$$
$$M_{t+1} := 0,$$
$$\beta_{t+1} := U_\pi^{i\to j}(s, a_t^s, \eta_t^j) - U_\pi^{i\to j}(s, a_t^s, \eta_\pi^{j,*}),$$

where we can see that the iteration algorithm

$$\nu_{t+1}^{i\to j}(s) = \nu_t^{i\to j}(s) + \alpha_t \cdot \left( h(\nu_t^{i\to j}(s), a_t^s) + M_{t+1} + \beta_{t+1} \right) \tag{22}$$

is equal to the iteration algorithm defined in Equation 17. Before showing the convergence of this iteration algorithm, we consider a different one. For this, observe that by Assumption 5.5, the sequence $\{\eta_t^j\}_{t\geq 0}$ is almost surely bounded. That means there exists $K > 0$ such that

$P(\sup_{t\geq 0}\left\|\eta_t^j\right\| < K) = 1$. Define the error term $\tilde{\beta}_t := \mathbb{I}_{\{\sup_{t\geq 0}\|\eta_t^j\|<K\}} \cdot \beta_t$, where $\mathbb{I}_A$ denotes the indicator function on a set $A$. Define the following iteration algorithm with the restricted error sequence $\tilde{\beta}_t$

$$\tilde{\nu}_{t+1}^{i\to j}(s) := \tilde{\nu}_t^{i\to j}(s) + \alpha_t \cdot \left(h(\tilde{\nu}_t^{i\to j}(s), a_t^s) + \tilde{\beta}_{t+1}\right). \tag{23}$$

To get the convergence of the iteration defined in Equation 23, we first check that part (a) to (e) from Assumption 3.3 hold. The function $h : \mathbb{R}^{k_j} \times \mathcal{A} \to \mathbb{R}$ is Lipschitz continuous in its first argument, as it is even linear in its first argument, i.e.,

$$|h(\nu, a) - h(\nu', a)| = |\nu - \nu'| \text{ for any } a \in \mathcal{A},$$

and part (a) holds. By Assumption 3.2, the Markov chain $\{a_t^s\}_{t\geq 0}$ is irreducible and aperiodic. Therefore, the Markov chain satisfies part (b). Furthermore, the stepsizes $\{\alpha_t\}_{t\geq 0}$ satisfy part (c). As the sequence $\{M_t\}_{t\geq 0}$ is the zero sequence, it is trivially a martingale difference sequence with a conditionally bounded norm, and satisfies part (d). Finally, for part (e), it remains to show that $\{\tilde{\beta}_t\}_{t\geq 0}$ is a bounded random sequence that converges to zero almost surely. For this, note that $\{\eta_t^j\}_{t\geq 0}$ is uniformly bounded on the set $\{\sup_{t\geq 0}\left\|\eta_t^j\right\| < K\}$. By Assumption 5.4, we get that the functions

$$\max_{a^i\in\mathcal{A}^i} Q_\pi^j(s, a^{-i}, a^i, \eta^j) \text{ and } \min_{a^i\in\mathcal{A}^i} Q_\pi^j(s, a^{-i}, a^i, \eta^j) \tag{24}$$

are continuous functions in $\eta^j$. Therefore, the function $U_\pi^{i\to j}(s, a_t^s, \cdot)$ is, as a sum of continuous functions, also continuous in $\eta^j$. In particular, as it is a continuous function on the compact set $\{\eta^j \in \mathbb{R}^{k_j} : \left\|\eta^j\right\| < K\}$, we get that it is bounded. Therefore, together with the convergence of $\eta_t^j \to \eta_\pi^{j,*}$, we get that $\{\tilde{\beta}_t\}_{t\geq 0}$ is a bounded random sequence that converges to zero. Therefore, Assumption 3.3 is satisfied for the iteration algorithm from Equation 23.

Next, consider the ODE given by

$$\dot{\nu}^{i\to j}(s) = \sum_{a\in\mathcal{A}} \pi(s, a) \cdot h\left(\nu^{i\to j}(s), a\right) = -\nu^{i\to j}(s) + \sum_{a\in\mathcal{A}} \pi(s, a) \cdot U_\pi^{i\to j}(s, a, \eta^{j,*}) \tag{25}$$

and define the right-hand side as $f(\nu^{i\to j}(s))$. We can see that $\nu^{i\to j}(s) = \sum_{a\in\mathcal{A}} \pi(s, a)U_\pi^{i\to j}(s, a, \eta^{j,*})$ is an equilibrium solution to the ODE above, and as $f$ is Lipschitz continuous, we get by the theorem of Picard-Lindelöf, see page 89 in the book of Adkins & Davidson (2012), that this solution is unique. Define the function $f_c(\nu^{i\to j}(s)) = c^{-1} \cdot f(c\nu^{i\to j}(s))$. Then $\lim_{c\to\infty} f_c(\nu^{i\to j}(s)) = -\nu^{i\to j}(s) =: f_\infty(\nu^{i\to j}(s))$ exists and the ODE $\dot{\nu}^{i\to j}(s) = f_\infty(\nu^{i\to j}(s))$ has the origin as unique asymptotically stable equilibrium. Therefore, we get by Theorem 3.5 that $\sup_{t\geq 0}\|\nu_t^{i\to j}(s)\| < \infty$ almost surely. Then, we can use Theorem 3.4 to conclude that

$$\tilde{\nu}_t^{i\to j}(s) \to SI^{i\to j}(s, \pi, \eta_\pi^{j,*}) \text{ almost surely.} \tag{26}$$

We extend this result to the original iteration sequence $\{\nu_t^{i\to j}(s)\}_{t\geq 0}$. As the sequence $\{\eta_t^j\}_{t\geq 0}$ is almost surely bounded, we see that

$$P\left(\left\{\omega \in \Omega : \sup_{t\geq 0}\left\|\eta_t^j(\omega)\right\| < K\right\}\right) = 1 \Leftrightarrow P\left(\left\{\omega \in \Omega : \sup_{t\geq 0}\left\|\eta_t^j(\omega)\right\| \geq K\right\}\right) = 0, \tag{27}$$

which means that $\left\{\sup_{t\geq 0}\left\|\eta_t^j\right\| \geq K\right\} = \left\{\sup_{t\geq 0}\left\|\eta_t^j\right\| < K\right\}^{\complement}$ is a null-set. Therefore, we get

$$\nu_{t+1}^{i\to j}(s) \to SI^{i\to j}\left(s, \pi, \eta^{j,*}\right) \text{ a.s.} \Leftrightarrow P\left(\left\{\omega \in \Omega : \lim_{t\to\infty}\nu_t^{i\to j}(s)(\omega) = SI^{i\to j}\left(s, \pi, \eta^{j,*}\right)(\omega)\right\}\right)$$

$$= P\left(\left\{\omega \in \Omega \setminus \left\{\sup_{t\geq 0}\left\|\eta_t^j\right\| < K\right\}^{\complement} : \lim_{t\to\infty}\nu_t^{i\to j}(s)(\omega) = SI^{i\to j}\left(s, \pi, \eta^{j,*}\right)(\omega)\right\}\right)$$

$$= P\left(\left\{\omega \in \Omega : \lim_{t\to\infty}\left(\nu_t^{i\to j}(s) \cdot \mathbb{I}_{\{\sup_{t\geq 0}\|\eta_t^j\|<K\}}\right)(\omega) = \left(SI^{i\to j}\left(s, \pi, \eta^{j,*}\right) \cdot \mathbb{I}_{\{\sup_{t\geq 0}\|\eta_t^j\|<K\}}\right)(\omega)\right\}\right)$$

$$= P\left(\left\{\omega \in \Omega : \lim_{t\to\infty}\tilde{\nu}_t^{i\to j}(s)(\omega) = SI^{i\to j}\left(s, \pi, \eta^{j,*}\right)(\omega)\right\}\right) = 1,$$

which gives us the statement. $\qquad\square$

A.5    PROOF FOR THEOREM 5.11 (CONTINUITY IN POLICY CHANGES)

To prove the continuity of SIM and TIM in the policy, we show the continuity of their individual terms. So, recall that, under Assumption 3.2, we can represent SIM and TIM of agent $i$ on agent $j$ by

$$SI^{i \to j}(s, \pi_\theta) = \sum_{a \in \mathcal{A}} \pi_\theta(s, a) \cdot U_\theta^{i \to j}(s, a), \qquad (28)$$

$$TI^{i \to j}(\pi_\theta) = \sum_{s \in \mathcal{S}} d_\theta(s) \sum_{a \in \mathcal{A}} \pi_\theta(s, a) \cdot U_\theta^{i \to j}(s, a). \qquad (29)$$

First, we show the continuity of the stationary distribution $d_\theta$ in $\theta \in \mathbb{R}^m$. To do this, we need some results from linear algebra, which we state without proof. The first is the well-known Perron-Frobenius theorem, which was originally introduced in this form by Frobenius (1912). Before stating the theorem, we introduce the notion of primitive matrices and a result connecting these to irreducible and aperiodic matrices. The following definition and two theorems are taken from the book of Seneta (2006).

**Definition A.1.** A square non-negative matrix $A$ is said to be primitive if there exists a positive integer $k$ such that $A^k > 0$.

The following theorem connects primitive to irreducible and aperiodic matrices. Note that transition matrices of irreducible and aperiodic Markov chains are irreducible and aperiodic.

**Theorem A.2.** A matrix $A$ is irreducible and aperiodic if and only if it is primitive.

With this, we now state the Perron-Frobenius theorem for primitive matrices.

**Theorem A.3.** Suppose $A$ is an $n \times n$ non-negative primitive matrix. Then there exists an eigenvalue $r$ such that:

    (a)  $r$ is a real value and strictly larger than $0$

    (b)  with $r$ can be associated strictly positive left and right eigenvectors

    (c)  $r > |\lambda|$ for any eigenvalue $\lambda \neq r$

    (d)  the eigenvectors associated with $r$ are unique to constant multiples

    (e)  $r$ is a simple root of the characteristic polynomial of $A$

One calls $r$ the Perron-Frobenius eigenvalue and its corresponding positive eigenvectors, the Perron-Frobenius eigenvectors.

The next theorem is an adapted version of a result of theorem 8 on page 130 in the book by Lax (2007), about the continuity of eigenvectors for simple eigenvalues.

**Theorem A.4.** Let $A(t)$ be a square matrix whose elements are continuously differentiable in $t \in \mathbb{R}^m$. Suppose that $a_0$ is an eigenvalue of $A(0)$ of multiplicity one, in the sense that $a_0$ is a simple root of the characteristic polynomial of $A(0)$. Then there exists a $\delta > 0$ such that for $\|t\| \leq \delta$, there exists an eigenvalue $a(t)$ of $A(t)$ that depends continuously differentiable on $t$, with $a(0) = a_0$. Furthermore, we can choose an eigenvector $h(t)$ of $A(t)$ pertaining to the eigenvalue $a(t)$ to depend continuously differentiable on $t$.

The original version of the theorem is for the case $m = 1$. However, the extension to multiple dimensions is straightforward, as one only needs to assume $A(t)$ to be continuously differentiable in $t \in \mathbb{R}^m$, instead of only differentiable and the proof carries over without changes. With the previous results, we now give our proof for the continuity of stationary distributions.

**Lemma A.5.** Let $A : \mathbb{R}^m \to \mathbb{R}^{n \times n}$ be a continuously differentiable function. Additionally, the matrix $A(t)$ is a transition matrix of an irreducible and aperiodic Markov chain for every $t \in \mathbb{R}^m$. Then the function $t \mapsto d(t)$ is continuously differentiable for all $t \in \mathbb{R}^m$, where $d(t) \in \mathbb{R}^n$ is the stationary distribution of $A(t)$.

*Proof.* As the matrix $A(t)$ is the transition matrix of an irreducible and aperiodic Markov chain, the associated stationary distribution $d(t)$ exists and is unique by theorem 4.1 on page 119 in the book of Seneta (2006). Therefore, the function $f : t \mapsto d(t)$ is well-defined. It remains to show that $f$ is continuously differentiable for all $t \in \mathbb{R}^m$.

For this, we want to use Theorem A.4 for the matrix $A(t)^T$, as the stationary distribution $d(t)$ is a right eigenvector of $A(t)^T$ to the eigenvalue one. We observe that $A(t)$ is a square non-negative irreducible and aperiodic matrix and by Theorem A.2 also primitive, i.e., there exists an $r \in \mathbb{N}$ such that $A(t)^r > 0$. Therefore, Theorem A.3 holds for $A(t)$. As $A(t)$ is row-stochastic, its largest eigenvalue is one and is by part (c) of Theorem A.3 also the Perron-Frobenius eigenvalue. We show next that the same holds for $A(t)^T$. For this, we observe that

$$A(t)^r > 0 \Leftrightarrow (A(t)^r)^T > 0 \Leftrightarrow \left(A(t)^T\right)^r > 0.$$

That means, $A(t)^T$ is primitive, i.e., the Perron-Frobenius theorem holds for $A(t)^T$ as well. Furthermore, for $\lambda \in \mathbb{R}$ it holds that

$$\det\left(A(t)^T - \lambda I\right) = \det\left((A(t) - \lambda I)^T\right) = \det\left(A(t) - \lambda I\right),$$

which means that $A(t)^T$ has the same eigenvalues as $A(t)$. This again means by part (c) of Theorem A.3 that the value one is also the Perron-Frobenius eigenvalue of $A(t)^T$. Using part (e) of Theorem A.3, we get that one is a simple root of the characteristic polynomial of $A(t)^T$.

Let $u \in \mathbb{R}^m$ be arbitrary. From our deductions about $A(t)^T$ so far, we can now use Theorem A.4. Therefore, there exists a $\delta > 0$ such that, on the set $\Lambda := \{t \in \mathbb{R}^m : \|t - u\|\}$, there exist continuously differentiable functions $a : \Lambda \to \mathbb{R}, h : \Lambda \to \mathbb{R}^n$. Whereas $a(t)$ is an eigenvalue of $A(t)^T$ with $a(u) = 1$ and $h(t)$ is an eigenvector of $A(t)^T$ pertaining to the eigenvalue $a(t)$.

The Perron-Frobenius Theorem holds for all $t \in \Lambda$ and therefore, by part (c) of Theorem A.3, we see for all eigenvalues $\lambda$ of $A(t)^T$ with $\lambda \neq 1$, that $|\lambda| < 1$. However, the function $a$ is continuously differentiable, which means that $a \equiv 1$. In turn, this means by part (b) that $h(t)$ is a strictly positive eigenvector for all $t \in \Lambda$. Now, define the scaling function

$$k(t) := \left(\sum_{i=1}^n h(t)_i\right)^{-1} \Leftrightarrow 1 = k(t) \cdot \left(\sum_{i=1}^n h(t)_i\right).$$

That means that $k(t) \cdot h(t)$ is an eigenvector of $A(t)^T$ to the eigenvalue one for all $t \in \Lambda$. In particular, that means that $k(t) \cdot h(t)$ is a stationary distribution of $A(t)$ and, therefore, $f(t) = k(t) \cdot h(t)$ for all $t \in \Lambda$. As $h(t)$ is strictly positive and continuously differentiable, $k$ is continuously differentiable as well. That means that $f$ is continuously differentiable for all $t \in \Lambda$, which means in particular that it is continuously differentiable in $u$. As $u \in \mathbb{R}^m$ has been arbitrary, we get that $f$ is continuously differentiable for all $t \in \mathbb{R}^m$. This concludes the proof. $\square$

Before we can prove the main result of this subsection, we need another lemma about the convergence of the expected distribution after several steps from an initial distribution of a Markov chain to its stationary distribution. The proof of this Lemma is inspired by the proof of the Convergence Theorem 4.9 on page 52 in the book of Levin et al. (2017).

**Lemma A.6.** Let $A : \mathbb{R}^m \to \mathbb{R}^{n \times n}$ be a matrix-valued continuously differentiable function and $\Lambda \subset \mathbb{R}^m$ a compact set. Furthermore, for every $t \in \mathbb{R}^m$, matrix $A(t)$ is a transition matrix of an irreducible and aperiodic Markov chain with stationary distribution $d(t)$. Finally, if there exists a $t_{i,j} \in \mathbb{R}^m$ such that the entry $A(t_{i,j})_{i,j} > 0$, then $A(t)_{i,j} > 0$ for all $t \in \mathbb{R}^m$. In this case, there exist constants $C \geq 1$ and $\alpha \in (0, 1)$ such that

$$\|x_0 A(t)^\tau - d(t)\|_2 \leq n \cdot C \alpha^\tau, \tag{30}$$

for all $t \in \Lambda$ and $j \in \{1, \ldots, n\}$ where $x_0$ is a non-negative vector, which entries sum up to one, that represents some initial distribution over the states of the Markov chain.

*Proof.* Since $A(t)$ is the transition matrix of an irreducible and aperiodic Markov chain it is primitive by Theorem A.2, i.e., there exists a minimal $r_t \in \mathbb{N}$ such that $A(t)^{r_t}$ has only positive entries. If an entry of $A(t)$ is positive, it is positive for all $t \in \mathbb{R}^m$. Together with the fact that $A(t)$ has no

negative entries for all $t \in \mathbb{R}^m$, we get that $r_t = r_u =: r$ for all $t, u \in \mathbb{R}^m$.

The entries of $A(t)$ are continuous on a compact set $\Lambda$, therefore there exists a minimal positive entry of $A(t)^r$ for all $t \in \Lambda$, which we denote by

$$a_{min} := \min\{(A(t)^r)_{i,j} \, | i, j \in \{1, \ldots n\}, t \in \Lambda\}.$$

By Lemma A.5, the entries of $d(t)$ are continuous for $t \in \Lambda$. Therefore, there exists a maximum entry of $d(t)$ for all $t \in \Lambda$, i.e.,

$$d_{max} := \max\{d(t)_i | i \in \{1, \ldots, n\}, t \in \Lambda\}.$$

Please note that $d(t)$ is a probability distribution that has only positive entries. Therefore, let $t \in \Lambda$ be arbitrary and take $\delta \in (0, 1)$ such that $a_{min} \geq \delta \cdot d_{max}$. Define $\lambda := 1 - \delta$ and let $D(t)$ be a matrix with $n$ rows, where all rows are equal to $d(t)$. Then the equation

$$A(t)^r = (1 - \lambda)D(t) + \lambda H(t), \tag{31}$$

defines a row-stochastic matrix $H(t)$. Note that, as every row of $D(t)$ is identical, $MD(t) = D(t)$ for every row-stochastic matrix $M \in \mathbb{R}^{n \times n}$. That means in particular that $H(t)D(t) = D(t)$ and $A(t)D(t) = D(t)$. The rows of $D(t)$ are the stationary distribution of $A(t)$, which means that $D(t)A(t) = D(t)$. Next, we show by induction that

$$A(t)^{rk} = \left(1 - \lambda^k\right) D(t) + \lambda^k H(t)^k, \text{ for } k \geq 1. \tag{32}$$

The starting condition for $k = 1$ holds by Equation 31. Assuming that it holds for $k = l$, we see

$$
\begin{aligned}
A(t)^{r(l+1)} &= A(t)^{rl} \cdot A(t)^r \\
&= \left[\left(1 - \lambda^l\right) D(t) + \lambda^l H(t)^l\right] \cdot A(t)^r \\
&= \left(1 - \lambda^l\right) D(t) \cdot A(t)^r + \lambda^l H(t)^l \cdot A(t)^r \\
&= \left(1 - \lambda^l\right) D(t) + \lambda^l H(t)^l \cdot \left[(1 - \lambda) D(t) + \lambda H(t)\right] \\
&= \left(1 - \lambda^l\right) D(t) + \lambda^l (1 - \lambda) H(t)^l \cdot D(t) + \lambda^{l+1} H(t)^{l+1} \\
&= \left(1 - \lambda^l\right) D(t) + \left(\lambda^l - \lambda^{l+1}\right) D(t) + \lambda^{l+1} H(t)^{l+1} \\
&= \left(1 - \lambda^{l+1}\right) D(t) + \lambda^{l+1} H(t)^{l+1},
\end{aligned}
$$

which shows that the claim in Equation 32 holds for all $k \geq 1$. Rearranging this equation yields

$$A(t)^{rk} - D(t) = \lambda^k \left(H(t)^k - D(t)\right). \tag{33}$$

For $j \in \mathbb{N}$, we multiply each side of Equation 33 by $A(t)^j$ and get

$$A(t)^{rk+j} - D(t) = \lambda^k \left(H(t)^k A(t)^j - D(t)\right). \tag{34}$$

Let $x_0 \in \mathbb{R}^n$ be a non-negative row-vector, which entries sum up to one. Multiplying Equation (34) by $x_0$ from the left and taking the $\|\cdot\|_2$-norm of both sides yields

$$\left\|x_0 A(t)^{rk+j} - d(t)\right\|_2 = \lambda^k \left\|x_0 H(t)^k A(t)^j - d(t)\right\|_2.$$

Note that $H(t)$ and $A(t)$ are row-stochastic matrices and that the product of row-stochastic matrices is again row-stochastic. Therefore, the rows of $H(t)^k A(t)^j$ sum up to one, as does a convex combination over the sum of the rows. Therefore, the distance to the stationary distribution $d(t)$ can be bounded by $n$. This gives us

$$\left\|x_0 A(t)^{rk+j} - d(t)\right\|_2 \leq n \cdot \lambda^k. \tag{35}$$

Now, define $C := \frac{1}{\lambda}$ and $\alpha := \lambda^{1/r}$. For $\tau > r$, there exist $k \in \mathbb{N}$ and $j \in \{1, \ldots, r\}$ such that $\tau = rk + j$. This gives us with Equation 35

$$\left\|x_0 A(t)^\tau - d(t)\right\|_2 \leq n \cdot \lambda^{\frac{\tau - j}{r}} = n \cdot \left(\frac{1}{\lambda}\right)^{\frac{j}{r}} \cdot \lambda^{\frac{\tau}{r}} \leq n \cdot C \cdot \alpha^\tau, \tag{36}$$

where we used that $C \geq 1$ and $j \leq r$ in the last step. For $\tau < r$, we note that $\lambda^{\frac{\tau}{r} - 1} \geq 1$ and as $A(t)$ is row-stochastic we get

$$\left\|x_0 A(t)^\tau - d(t)\right\|_2 \leq n \leq n \cdot \lambda^{\frac{\tau}{r} - 1} = n \cdot C \cdot \alpha^\tau. \tag{37}$$

As $t \in \Lambda$ has been arbitrary and the constants $C$ and $\alpha$ are independent of $t$, Equations 36 and 37 conclude the statement. $\square$

With this result, we can establish the continuity of SIM and TIM in changes in the policy.

**Theorem A.7.** Let $\Theta \subset \mathbb{R}^m$ be a compact set, and let $\pi_\theta$ be the joint policy. Under Assumptions 3.2 and 5.10, the total impact measurement $TI^{i \to j}(\pi_\theta)$ and state impact measurement $SI^{i \to j}(s, \pi_\theta)$ are continuous in $\theta \in \Theta$ for every $i, j \in \mathcal{N}$ and $s \in \mathcal{S}$.

*Proof.* Under Assumption 3.2, one can represent SIM and TIM using Equations 28 and 29. From this, we see that the continuity in $\theta \in \Theta$ follows if we can show continuity of the individual terms. According to Assumption 5.10, the term $\pi_\theta(s, a)$ is continuous in $\theta$ for every $s \in \mathcal{S}$ and $a \in \mathcal{A}$. Additionally, this means that the function $\theta \mapsto P_\theta$ is continuously differentiable as well, and denotes the transition matrix of the irreducible and aperiodic Markov chain over the states $\{s_t\}_{t \geq 0}$ for every $\theta \in \mathbb{R}^m$. By using Lemma A.5, we get that the stationary distribution $d_\theta$ is continuous in $\theta \in \Theta$. Therefore, it remains to show that $U_\theta^{i \to j}(s, a)$ is continuous in $\theta$. The impact sample for state $s$ and action $a$ is given by

$$U_\theta^{i \to j}(s, a) = \max_{a^i \in \mathcal{A}^i} Q_\theta^j(s, a^{-i}, a^i) - \min_{a^i \in \mathcal{A}^i} Q_\theta^j(s, a^{-i}, a^i).$$

Assume $Q_\theta^j(s, a)$ is continuous in $\theta \in \Theta$, then we get that $U_\theta^{i \to j}(s, a)$ is continuous in $\theta \in \Theta$ for every $i, j \in \mathcal{N}, s \in \mathcal{S}$, and $a \in \mathcal{A}$ as maximum or minimum of finitely many continuous terms. To complete the proof, it remains to show the continuity of $Q_\theta^j(s, a)$ in $\theta \in \Theta$.

Denote the transition matrix of the Markov chain $\{s_t, a_t\}_{t \geq 0}$ induced by the policy $\pi_\theta$ by $P_\theta^{\mathcal{A}}$. The stationary distribution of $\{s_t, a_t\}_{t \geq 0}$ is given by $d_\theta^{\mathcal{A}} = \{d_\theta(s) \cdot \pi_\theta(s, a)\}_{s \in \mathcal{S}, a \in \mathcal{A}} \in \mathbb{R}^{|\mathcal{S} \times \mathcal{A}|}$. Furthermore, for a state-action pair $(s, a)$ set $v_0$ as starting distribution, where the entry corresponding to $(s, a)$ equals one and zero else and denote the expected rewards vector of agent $j$ by $R^j = \{R^j(s, a)\}_{s \in \mathcal{S}, a \in \mathcal{A}}$. Then the state-action function can be written as

$$Q_\theta^j(s, a) = \sum_{t \geq 0} \left( v_0 \left( P_\theta^{\mathcal{A}} \right)^t - d_\theta^{\mathcal{A}} \right)^T R^j := f(\theta).$$

Additionally define a sequence of functions $\{f_t\}_{t \geq 0}$ as

$$f_t(\theta) := \sum_{\tau=0}^{t} \left( v_0 \left( P_\theta^{\mathcal{A}} \right)^\tau - d_\theta^{\mathcal{A}} \right)^T R^j.$$

Observe that $\theta \mapsto P_\theta^{\mathcal{A}}$ is a continuously differentiable matrix-valued function by Assumption 5.10. Furthermore, it is a transition matrix of an irreducible and aperiodic Markov chain over the states and actions $\{s_t, a_t\}_{t \geq 0}$. Finally, as the transition probabilities of the underlying MDP $P(s'|s, a)$ are constant, and the probabilities of the policy satisfy $\pi(s, a) > 0$ for all $s, s' \in \mathcal{S}, a \in \mathcal{A}$, entries of $P_\theta^{\mathcal{A}}$ are positive for all $\theta \in \mathbb{R}^m$ if they are positive for one $\theta \in \mathbb{R}^m$. That means we can apply Lemma A.6 and get that there exist constants $C \geq 1$ and $\alpha \in (0, 1)$ such that $\left\| v_0 \left( P_\theta^{\mathcal{A}} \right)^t - d_\theta^{\mathcal{A}} \right\|_2 < |\mathcal{S} \times \mathcal{A}| \cdot C \alpha^t$ for every $t \geq 0$. Then we see that with

$$
\begin{aligned}
\sup_{\theta \in \Theta} |f(\theta) - f_t(\theta)| &= \sup_{\theta \in \Theta} \left| \sum_{\tau \geq t+1} \left( v_0 \left( P_\theta^{\mathcal{A}} \right)^\tau - d_\theta^{\mathcal{A}} \right)^T R^j \right| \\
&\leq \sup_{\theta \in \Theta} \sum_{\tau \geq t+1} \left| \langle v_0 \left( P_\theta^{\mathcal{A}} \right)^\tau - d_\theta^{\mathcal{A}}, R^j \rangle \right| \\
&\leq \sup_{\theta \in \Theta} \sum_{\tau \geq t+1} \left\| v_0 \left( P_\theta^{\mathcal{A}} \right)^\tau - d_\theta^{\mathcal{A}} \right\|_2 \cdot \left\| R^j \right\|_2 \\
&\leq \left\| R^j \right\|_2 \cdot |\mathcal{S} \times \mathcal{A}| \cdot C \cdot \sum_{\tau \geq t+1} \alpha^\tau,
\end{aligned}
$$

we can bound the difference of $f$ and $f_t$ uniformly for all $\theta \in \Theta$. As $|\alpha| < 1$ we get that

$$\lim_{t \to \infty} \sup_{\theta \in \Theta} |f(\theta) - f_t(\theta)| = 0.$$

Therefore, the sequence $\{f_t\}_{t \geq 0}$ converges uniformly on $\Theta$ to $f$. As the function $f_t$ is a finite sum of continuous functions, it is continuous in $\theta \in \Theta$ for every $t \geq 0$. By the uniform limit theorem (Forster, 2013), we get that $f$ is continuous as well, which concludes the proof. $\square$

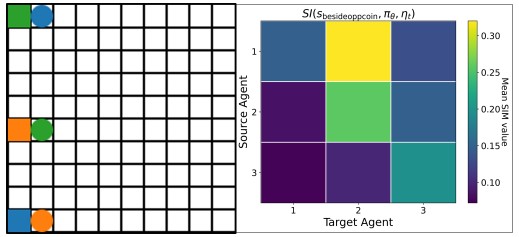 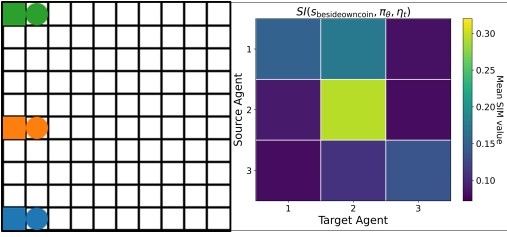

(a) SIM impact matrix $SI(s_{\text{beside opponent coin}}, \pi_\theta, \eta_t)$.   (b) SIM impact matrix $SI(s_{\text{beside own coin}}, \pi_\theta, \eta_t)$

Figure 3: Average SIM estimation over 10 seeds for specific grid positions. Once agents spawn beside their opponent's coin and once beside their own coin.

## B   SIM RESULTS IN COIN GAME

In this section, we provide additional findings related to the SIM estimation within the context of the coin game. Specifically, we assess the descriptive capabilities of SIM by closely examining particular states.

Considering the coin game's unique reward structure, we have spotlighted two distinct grid positions for our evaluation. Note that while it is possible, the likelihood of these board positions arising during training is low, which shows also the generalization capabilities of the SIM estimation. Nevertheless, we found that training the SIM estimations suffer from a higher variance than the TIM estimations. Which is why we restrict ourselves rather to a relative comparison between the two states instead of directly interpreting the absolute values.

The first scenario, illustrated in Figure 3a, situates the agents directly adjacent to an opponent's coin. To detail further, agent 1 is positioned next to agent 2's coin, agent 2 is adjacent to agent 3's coin, and agent 3 finds itself beside agent 1's coin. If, in their subsequent move, all agents opt to collect these coins, each will earn a reward of 1. Beyond this, agent 1 would impose a penalty on agent 2, while agent 2 would similarly penalize agent 3. Agent 3 cannot inflict penalties on any of the others.

In our second scenario, presented in Figure 3b, all agents are spawned directly beside their respective coins. In this configuration, there are no penalties in the immediate next time-step. Thus, even though penalization could arise in later iterations, the expected impact on agents that are susceptible to penalties should be considerably diminished when compared to the first scenario.

We compare the estimated state impact measurements in the considered scenarios, which are illustrated on the right side of Figures 3a and 3b, next. When assessing agent 1, we observe that its estimated impact on agent 2 is markedly high when poised to collect the latter's coin. However, this impact drops significantly when agent 1 is about to retrieve its own coin. Agent 2 showcases a different pattern. In the penalty-rich first scenario, its estimated impact is high for both itself and agent 3. However, when there's no looming penalty—as when agent 2 targets its own coin—the projected impact on agent 3 drops significantly.

Drawing from these observations, it becomes evident that the state impact measurements align with intuitive expectations based on the coin game's unique reward structure. The SIM's responsiveness to agent positioning, in relation to coin ownership and the subsequent potential for penalties, showcases its efficacy and descriptive power. In essence, this analysis demonstrates the robustness of SIM and its applicability as a tool for understanding agent interdependencies and influence in complex multi-agent environments.

## C   DETAILED EXPERIMENT SETUP

We give a detailed overview of the setup and methodology of the empirical experiments. We repeat the already mentioned parts for the convenience of the reader.

## C.1 RANDOM ENVIRONMENT

The specifics of the generation process are taken from the work of Zhang et al. (2018). Consider a set of $|\mathcal{N}| = 5$ agents and $|\mathcal{S}| = 5$ states. Each agent can choose from a binary action space $\mathcal{A}^j = \{0, 1\}$. We uniformly sample the values of the transition probabilities $P(s'|s, a)$ from the interval $[0, 1]$ and store them in a $|\mathcal{S}| \times |\mathcal{S}| \cdot |\mathcal{A}|$ matrix. To ensure irreducibility and aperiodicity of the resulting Markov chain over the states, we add the constant $5\mathrm{e}{-}5$ to all entries and normalize the rows of the matrix, so that they sum up to one. We sample the reward matrix's entries uniformly from $[0, 2]$ and store them in a $|\mathcal{N}| \times |\mathcal{S}| \cdot |\mathcal{A}|$ matrix. That means for every agent $j$, state $s \in \mathcal{S}$, and action $a \in \mathcal{A}$ a reward matrix entry $R^j(s, a)$ is sampled uniformly from $[0, 2]$. Furthermore, we create embeddings for the state and action pairs $\phi(s, a^j) \in \mathbb{R}^{m_j}$, with $m_j = 20$ for all $j \in \mathcal{N}$, by sampling every entry uniformly from $[0, 1]$. Accordingly, we sample the entries of the policy parameters $\theta^j \in \mathbb{R}^{m_j}$ uniformly from $[-\frac{1}{\sqrt{m_j}}, \frac{1}{\sqrt{m_j}}]$. To determine the probability distribution of the policies $\pi_{\theta^j}^j$, we use the Boltzmann policies (Sutton & Barto, 2018), i.e.,

$$\pi_{\theta^j}^j(s, a^j) = \frac{\exp\left(\phi(s, a^j)^T \theta^j\right)}{\sum_{b_j \in \mathcal{A}^j} \exp\left(\phi(s, b^j)^T \theta^j\right)}.$$

At time step $t$, where the system is in state $s_t$, the actions $a_t = (a_t^1, \ldots a_t^N)$ are sampled according to $a_t^j \sim \pi_{\theta^j}^j(s_t|\cdot)$. The instantaneous reward of agent $j$ is then given by $R^j(s_t, a_t)$. The following state $s_{t+1}$ is sampled from $P(\cdot|s_t, a_t)$, and the initial state $s_0$ is sampled uniformly from $\mathcal{S}$. Note that this setup satisfies Assumption 3.2 and the policies satisfy Assumption 5.10.

If we conduct experiments with some enforced influence structure among the agents, we adjust the environment in the following way. We set the rows of the transition probability matrix all equal to the first row, i.e., $P(s'|\cdot, \cdot) = P(s^0|\cdot, \cdot)$ for all $s' \in \mathcal{S}$. Additionally, we set the entries for a state $s \in \mathcal{S}$, and actions $a^{-i} = (a^1, \ldots, a^{i-1}, a^{i+1}, \ldots a^N) \in \mathcal{A}^{-i}$ in the reward matrix to $R^j(s, a^{-i}, a^i) = R^j(s, a^{-i}, \hat{a}^i)$ for all $a^i, \hat{a}^i \in \mathcal{A}^i$.

For all experiments, we assume that each agent can at least influence its individual state-action function. To determine the overall dependency structures among the agents, we randomly sample a number of additional dependencies $L_{\mathrm{add}}$ from the remaining ones. For example, for the dependency structure with $L_{\mathrm{add}} = 1$, each agent can have an influence on its state-action function and there exists exactly one pair of agents $i, j \in \mathcal{N}$ with $i \neq j$ such that $i$ can influence the individual state-action function of $j$. Enforcing no influence structure corresponds to the case of $L_{\mathrm{add}} = N^2 - N = 20$.

We determine different quantities analytically in the following way. The stationary distribution over the states and actions $d_\theta^{\mathcal{A}}$ is the left eigenvector to the eigenvalue one of the transition matrix $P_\theta^{\mathcal{A}}$. It can be easily determined by solving the linear system $\left(P_\theta^{\mathcal{A}}\right)^T - I_{|\mathcal{S} \times \mathcal{A}|} = 0$ and extracting a strictly positive vector of length one. The individual long-term return of agent $j$ is given by $J^j(\pi_\theta) = d_\theta^{\mathcal{A}} R^j$. Under Assumption 3.2, we can express the individual state-action functions as

$$Q_\theta^j(s, a) = \sum_{t \geq 0} \left(x_0 \left(P_\theta^{\mathcal{A}}\right)^t - d_\theta^{\mathcal{A}}\right)^T R^j,$$

where $x_0 \in \mathbb{R}^{1 \times |\mathcal{S}| \cdot |\mathcal{A}|}$ is the starting distribution with $x_0(s, a) = 1$ and zero else. By using $\left\|x_0 \left(P_\theta^{\mathcal{A}}\right)^t - d_\theta^{\mathcal{A}}\right\| < 1\mathrm{e}{-}8$ as stopping criterion, we can approximate the individual state-action functions analytically. We denote the individual state-action function approximations that use this method with $Q_\theta^{j,\mathrm{ana}}$ for $j \in \mathcal{N}$. Finally, we can use the analytically approximated individual state-action functions $Q_\theta^{j,\mathrm{ana}}$ to approximate SIM and TIM analytically. For this, we use the formulas from Equations 28 and 29. The corresponding SIM and TIM matrices, with entries calculated by this method, are denoted by $TI^{\mathrm{ana}}(\pi_\theta)$ and $SI^{\mathrm{ana}}(s, \pi_\theta)$ respectively.

For a learning approximation of individual state-action functions $Q_\theta^j(\cdot, \cdot, \eta^j)$, we use the tabular SARSA algorithm for the average-reward setting, see Section 10.3 in the book of Sutton & Barto (2018). The learning rates $\alpha$ and $\beta$ are set to $\alpha = \beta = 0.036$. The entries of the state-action table are initially set to one. The TIM and SIM approximation matrices using the learning state-action function for the iteration algorithm are denoted by $TI(\pi_\theta, \eta_t)$ and $SI(s, \pi_\theta, \eta_t)$.

For both approximation algorithms, we initialize the approximation of SIM and TIM for all $i, j \in \mathcal{N}$ to $\frac{1}{|\mathcal{N}|} = 1/5$ and set the learning rates to $\alpha_t^{\text{TIM}} = \frac{0.471}{t^{0.726}}$ and $\alpha_t^{\text{SIM}} = \frac{0.74}{t^{0.539}}$.

## C.2   COIN GAME

The coin game is structured within a $10 \times 10$ grid where three agents initially start at random positions, ensuring no overlaps. For every state $s$, agents have five potential actions: moving left, right, up, down, or staying put. Movements are constrained by the grid's boundaries—if an agent attempts to exit the grid, it remains stationary. When two agents simultaneously aim for the same spot, one's move is prioritized at random, leaving the other unmoved.

Each agent is associated with a distinct coin type. Coins spawn at random time intervals on random positions, avoiding any overlap with existing coins. If a coin materializes on an agent-occupied spot, the agent must stay there for an additional round to claim it. However, an agent collects a coin when it moves on a field that is already occupied by a coin. The spawn rate is chosen so that, in expectation, a coin of each type appears every four to five steps, with a cap of four coins of the same type coexisting on the grid. The average collecting rate of the agents with a trained (greedy) policy, is slightly below the expected spawning rate, so that there usually are coins present on the field.

The reward dynamics of the coin game are structured in the following manner. Upon collecting any coin, an agent is awarded a reward of 1, irrespective of the coin's type. However, if an agent collects a coin not designated for it, the owner of that coin might incur a penalty. The penalty system is deliberately one-sided. Agent 3 is penalized by $-2$ each time an opposing agent collects one of its coins. Agent 2 specifically incurs a penalty when Agent 1 collects one of its coins. This means Agent 1 can consistently impose penalties by collecting opposing coins, while Agent 3 never penalizes opponents through its actions. It's noteworthy that, while the standard coin game is constructed as a social dilemma, our adaptation deviates from this by ensuring agents 1 and 2 faces no adverse consequences from acting greedily.

The representation $\phi(s) \in \mathbb{R}^{6 \times 10 \times 10}$ of a state $s$ is encoded in a one-hot tensor with dimensions $6 \times 10 \times 10$, where six channels detail the game's state. Specifically, each agent occupies two channels: one marking its location and another indicating the placements of its designated coins. All agents can observe the full state $s$.

We examine this environment with evolving acting policies. Each agent $j$ operates under an independent policy $\pi_{\theta_t^j}^j$. The parameters $\theta_t^j$ are refined using the Proximal Policy Optimization (PPO) algorithm (Schulman et al., 2017). Despite our focus on a continuing task, we implement the conventional PPO approach, targeting the maximization of the discounted cumulative reward. This choice underscores the versatility and independence of TIM and SIM from acting policies, given that they are formulated for the average reward setting.

For estimating the on-policy state-action functions, denoted as $Q_{\theta_t}^j : \mathbb{R}^{6 \times 10 \times 10} \times \mathcal{A} \to \mathbb{R}$, each agent employs a deep SARSA network (Zhao et al., 2016). We have tailored the network's update rule to provide estimates for the state-action values consistent with the average reward setting, as defined by Equation 2.

We initialize the approximation of TIM for all $i, j \in \mathcal{N}$ to $\frac{1}{|\mathcal{N}|} = 1/3$. Rather than adjusting the TIM estimates at every incremental time step, we accumulate a batch of transitions and then update the estimate, using the mean impact samples derived from this batch. We found that this approach is more stable. Importantly, this modification does not conflict with the theoretical guarantees associated with the original update rule as denoted in Equation 12.

Due to the large state space, the tabular SIM estimation approach delineated in Section 5.2 becomes infeasible. To circumvent this limitation, for every agent $i$, we introduce an additional neural network featuring three output heads—one for each agent—to estimate $SI^{i \to j}(\phi(s), \pi_{\theta_t})$. We proceed by training this network using a supervised method. Specifically, we gather a batch of transitions, identify the associated impact samples, and then condition the network to align with these impact samples. To ensure a more stable and consistent training process, transitions are stored in a replay buffer, from which batches are subsequently drawn.

For implementing the PPO algorithm, we leverage the standard version offered by StableBaselines3 (SB3) (Raffin et al., 2021). The deep SARSA algorithm can be conceptualized as an on-policy version of the widely recognized DQN algorithm (Mnih et al., 2013). Accordingly, we modify the DQN implementation of SB3 to the deep SARSA algorithm for the average-reward setting. It is worth noting that the SB3 library is traditionally designed for single-agent scenarios. In our case, we have modified it to ensure concurrent learning across all our algorithms. In terms of computational strategy, we chose to run the environment simulations entirely on a GPU. This approach allows us to run $10,000$ environments in parallel. Consequently, a single timestep, denoted as $t$, produces a batch of $10,000$ transitions for us. All of our experiments specific to the coin game were conducted using a consumer-grade Nvidia Geforce RTX 2080Ti GPU.

# D  HYPERPARAMETERS FOR EXPERIMENTS

In this section, we give a detailed overview of the used hyperparameters and how they were chosen for the empirical experiments.

## D.1  RANDOM ENVIRONMENT

A summary of the used hyperparameters are given in Table 1.

Table 1: Overview hyperparameters in random environment determined by random search.

| Declaration | Symbol | Algorithm | Value |
|---|---|---|---|
| State-action learning rate | $\alpha$ | SARSA algorithm | 0.036 |
| Long-term return step size | $\beta$ | SARSA algorithm | 0.036 |
| Initial Learning rate (TIM) | $\alpha_0^{\text{TIM}}$ | TIM approximation | 0.471 |
| Learning rate decay (TIM) | $d_{\text{decay}}^{\text{TIM}}$ | TIM approximation | 0.726 |
| Initial Learning rate (SIM) | $\alpha_0^{\text{SIM}}$ | SIM approximation | 0.740 |
| Learning rate decay (SIM) | $d_{\text{decay}}^{\text{SIM}}$ | SIM approximation | 0.539 |
| Critic initial learning rate | $\beta_{0,\omega}$ | Algorithm 1 of Zhang et al. | 0.128 |
| Critic learning rate decay | $d_{\omega,\text{decay}}$ | Algorithm 1 of Zhang et al. | 0.039 |
| Actor initial learning rate | $\beta_{0,\theta}$ | Algorithm 1 of Zhang et al. | 0.924 |
| Actor learning rate decay | $d_{\theta,\text{decay}}$ | Algorithm 1 of Zhang et al. | 0.088 |
| Size of state individual action embedding | $m$ | Algorithm 1 of Zhang et al. | 20 |
| Size of state joint action embedding | $K$ | Algorithm 1 of Zhang et al. | 80 |

The experiments to evaluate the approximation algorithms for SIM and TIM have several tuneable hyperparameters. First, we determined the learnings rates $\alpha$ and $\beta$ for the SARSA approximation algorithm (Sutton & Barto, 2018), the initial learning rates $\alpha_0^{\text{SIM}}$ and $\alpha_0^{\text{TIM}}$, and the decay rates $d_{\text{decay}}^{\text{SIM}}$ and $d_{\text{decay}}^{\text{TIM}}$ of the TIM approximation algorithms. The learning rates for the SIM and TIM approximation algorithms in timestep $t$ were then given by

$$\alpha_t^{\text{SIM}} = \frac{\alpha_0^{\text{SIM}}}{t^{d_{\text{decay}}^{\text{SIM}}}} \text{ and } \alpha_t^{\text{TIM}} = \frac{\alpha_0^{\text{TIM}}}{t^{d_{\text{decay}}^{\text{TIM}}}}.$$

Next, we set the number of agents $N = 5$, the number of states $|\mathcal{S}| = 5$, the size of the action spaces $\left|\mathcal{A}^j\right| = 2$ for all $j \in \mathcal{N}$, the number of approximation steps to $T = 10.000$, initialized the tables of the individual state-action function approximations with one, initialized the approximation of SIM and TIM by $1/|\mathcal{N}| = 1/5$, and set $\alpha = \beta$. With this in place, we performed a random search with $1.000$ different seeds and sampled $\alpha, \alpha_0$, and $d_{\text{decay}}$ uniformly from $[0, 1]$ for every seed. The remainder of the environment and the policy parameters were randomly chosen without enforcing any influence structure. The details of this can be found in Section C. We then measured the error of the approximation algorithms using the SARSA algorithm to the analytically determined SIM and TIM for a given policy, i.e., we measured $\|SI^{\text{ana}}(\cdot, \pi_\theta) - SI(\cdot, \pi_\theta, \eta_t)\|_1$ and $\|TI^{\text{ana}}(\pi_\theta) - TI(\pi_\theta, \eta_t)\|_1$ after $T$ steps. We chose the set of parameters that resulted in the minimal approximation errors over the random search. This gave us $\alpha = \beta = 0.036$, $\alpha_0^{\text{SIM}} = 0.740$, $\alpha_0^{\text{TIM}} = 0.471$, $d_{\text{decay}}^{\text{SIM}} = 0.539$, and $d_{\text{decay}}^{\text{TIM}} = 0.726$.

The hyperparameters for the actor-critic algorithm of Zhang et al. (2018) were determined similarly. For this we set the number of agents to $N = 10$, the number of states to $|\mathcal{S}| = 10$, the size of the action-spaces $|\mathcal{A}^j| = 2$ for all $j \in \mathcal{N}$, the number of simulation steps to $T = 20.000$, and do not enforce any influence structure. We perform a random search for $1.000$ different seeds. For each simulation, we sample the initial learning rate for the critic $\beta_{\omega,0}$ and actor $\beta_{\theta,0}$ uniformly from $[0, 1]$. The corresponding decay rate for the critic $d_{\omega,\text{decay}}$ is uniformly sampled from $[0, 1]$, and the decay rate for the actor $d_{\theta,\text{decay}}$ is sampled from $[d_{\omega,\text{decay}}, 1]$. This results in learnings rates at time step $t$ given by

$$\beta_{t,\omega} = \frac{\beta_{0,\omega}}{t^{d_{\omega,\text{decay}}}} \text{ and } \beta_{t,\theta} = \frac{\beta_{0,\theta}}{t^{d_{\theta,\text{decay}}}}.$$

Furthermore, we sample the size $m = m_j$ for every $j \in \mathcal{N}$ of the embeddings of the states and individual actions uniformly from the set $\{5, 10, 15, 20, 25, 30, 35, 40, 45, 50\}$, and the size of the embeddings of the state and joint actions denoted by $K$ uniformly from $\{10, 20, 30, 40, 50, 60, 70, 80, 90, 100\}$. The remainder of the environment and parameters, e.g., for the policies or global state-action function approximations, were chosen randomly. For each run, we measure the globally averaged long-term return, which is given for time step $t$ by

$$J_t = \frac{1}{t} \sum_{k=0}^{t-1} \frac{1}{N} \sum_{j \in \mathcal{N}} r_{k+1}^j, \tag{38}$$

where $r_{k+1}^j$ denotes the instantaneous reward of agent $j$ in time step $k$. We chose the final parameters from the simulation with the highest final globally averaged long-term return, i.e., the parameter set with the highest value of $J_{20.000}$ of the simulations. These parameters were $\beta_{\omega,0} = 0.128$, $\beta_{\theta,0} = 0.924$, $d_{\omega,\text{decay}} = 0.039$, $d_{\theta,\text{decay}} = 0.088$, $m = 20$, and $K = 80$.

The performance of the actor-critic learning algorithm, i.e., the value $J_t$, is sensitive to the parameter choice. The learned policies do not perform better than random for most of the randomly sampled parameters.

## D.2 COIN GAME

Given the coin game's complexity and the intricacies of the algorithms we employ, a random search approach—like the one used in the random environment—is infeasible. Instead, we opt for hand-tuning the hyperparameters, building upon the default settings provided by SB3. Modifications from these default values are detailed in Table 2.

All neural networks utilized in our study employ a consistent architecture for feature extraction. Specifically, the feature extractor consists of three convolutional layers, each with a kernel size of three and a stride of one, which is subsequently followed by a ReLU activation function. For the networks dedicated to the PPO and deep SARSA algorithms, the remaining architecture builds upon SB3's default networks, which operate on the embedded output from the feature extractor. On the other hand, the SIM network uses a simple feed forward design, comprised of three fully connected layers. Each of these layers has a hidden size of $64$ and is followed by ReLU activations.

Table 2: Overview hyperparameters in coin game determined by hand tuning.

| Declaration | SB3 variable name or symbol | Algorithm | Value |
|---|---|---|---|
| State-action learning rate | learning_rate | Deep SARSA | 0.0001 |
| Replay buffer size | buffer_size | Deep SARSA | $500,000$ |
| Transitions before training starts | learning_starts | Deep SARSA | $100,000$ |
| Target network update interval | target_update_interval | Deep SARSA | $500,000$ |
| Training batch size | batch_size | Deep SARSA | 8096 |
| Number of steps $t$ before update | n_rollout_steps | Deep SARSA | 8 |
| Long-term return step size | - | Deep SARSA | 0.01 |
| Initial Learning rate (TIM) | $\alpha_0^{\text{TIM}}$ | TIM approximation | 1.0 |
| Learning rate decay (TIM) | $d_{\text{decay}}^{\text{TIM}}$ | TIM approximation | 0.50 |
| Learning rate (SIM) | - | SIM approximation | 0.0001 |
| Discount rate | $\gamma$ | PPO | 0.98 |
| Learning rate | learning_rate | PPO | 0.001 |
| Number of steps $t$ before update | n_rollout_steps | PPO | 25 |

