# OpenReview forum: "Detecting Influence Structures in Multi-Agent Reinforcement Learning"
_ICLR.cc/2024/Conference — Submitted to ICLR 2024_

### Official Review · Reviewer_qZgK · 2023-10-30

**Soundness:** 2 fair
**Presentation:** 2 fair
**Contribution:** 3 good
**Rating:** 5
**Confidence:** 2

**Summary:**

The paper delves into the exploration of influence structures within Multi-Agent Reinforcement Learning (MARL) systems. The authors introduce influence measurement functions, serving as a comprehensive descriptive framework for understanding influence structures across various common settings. Within this framework, they introduce both total and state influence measures, specifically tailored to the context of average reward settings. The paper goes on to offer a rigorous theoretical analysis, focusing on the stability of these measures, as well as providing insights into the convergence properties and error bounds of the corresponding approximation algorithms. In addition to theoretical contributions, the authors conduct experiments in randomly generated environments to showcase the convergence of approximation errors, even in scenarios involving evolving policies. Moreover, the "coin game" experiment is presented as an example, demonstrating the practical applicability of these concepts in complex and dynamic settings. This experiment sheds light on the understanding of influence within black-box environments. Overall, this work sheds valuable light on the intricate world of influence structures in MARL systems, offering both theoretical insights and practical applicability in diverse and dynamic scenarios.

**Strengths:**

1. The problem this paper considers is rather important.
2. The literature review is sufficient.
3. The proposed framework seems to be novel.
4. The theoretical results are sound.

**Weaknesses:**

1. The evaluation of the proposed method is not enough. Figure 1 and Figure 2 only show the TIM error and mean TIM approximations in two environments respectively. What about the overall performance improvement of the proposed method?
2. This paper did not show any empirical comparison between the proposed method and baselines.
3. This paper did not provide any limitation discussions of the proposed method.

**Questions:**

1. How do TIM and SIM combine with MARL methods?
2. What is the main difference between TIM and SIM?

---

> ### Author Response · Authors · 2023-11-20
> **TIM and SIM within MARL**
>
> Thank you for your review and for highlighting the key aspects of our work. We appreciate the opportunity to respond to your queries and address the points of concern you have raised.
>
> ## Response to Questions
>
> ### How do TIM and SIM Combine with MARL Methods?
> TIM and SIM can be effectively combined with MARL methods in two significant ways. First, TIM and SIM serve as descriptive tools to assess interdependencies among agents in a given system. These measurements are based on observed actions, states, and rewards, enabling their combination with any MARL algorithm. In the coin game experiment, for instance, agents were trained using independent Proximal Policy Optimization (PPO) policies while we simultaneously measured their interdependencies using TIM and SIM. However, we could also have used a cooperative multi-agent learning algorithm such as QMIX [1] to provide insights into which agents exert the most influence under this learning algorithm. TIM and SIM enable this without prior knowledge of the environment.
> Secondly, beyond descriptive analysis, TIM and SIM can be employed to enhance the learning of joint policies. For example, SIM can be used as an intrinsic reward signal to prioritize states with higher impact, which are strategically more relevant [2]. Similarly, TIM can facilitate the construction of an approximate coordination graph [3]. However, our paper focuses on the first application, leaving the exploration of these applications for future work.
> ### Main Difference Between TIM and SIM
> TIM and SIM differ in their granularity of capturing interdependencies. SIM provides a state-specific impact analysis. It quantifies the expected impact one agent has on another in a particular state. For example, in different scenarios within the coin game, SIM would reflect varying impacts based on the proximity of agents to coins, capturing nuances in influence that change with the state context.
> TIM, on the other hand, represents the expected value of SIM over the stationary distribution of states under the current joint policy. It provides a more holistic view of influence across all states, rather than the state-specific analysis offered by SIM.
>
> ## Addressing raised weaknesses
> ### Evaluation and Comparison with Baselines
> Our evaluation, as depicted in Figures 1 and 2, focuses on demonstrating the practical effectiveness of the algorithms for measuring TIM and SIM. While we acknowledge that a direct comparison to baselines was not provided, this is primarily because there are no existing baselines directly comparable to our approach.
> The empirical results validate the practical functionality of our proposed algorithms. The small setting showcases the accuracy of our measurements compared to true values, and the coin game setting demonstrates that TIM and SIM can be reliably estimated in complex scenarios with meaningful semantic implications.
>
> We hope this response addresses your concerns and clarifies the contributions and potential applications of our work in the field of Multi-Agent Reinforcement Learning. Thank you once again for your valuable feedback and for providing us with the opportunity to enhance our research.
>
> ### Bibliography:
>
> [1]: Rashid et al., QMIX: Monotonic value function factorisation for deep multi-agent reinforcement learning, ICML2018
>
> [2]: Tonghan Wang, Jianhao Wang, Yi Wu, Chongjie Zhang, Influence-based multi-agent exploration, ICLR2020
>
> [3]: Carlos Guestrin, Shobha Venkataraman, and Daphne Koller. Context-Specific Multiagent Coordination and Planning with Factored MDPs, ICML2002

---

> > ### Author Response · Authors · 2023-11-22
> > **End of discussion period**
> >
> > In light of the end of the author-reviewer discussion period, we would again highlight our response. We hope that we adequately addressed your concerns and convinced you to change your score accordingly.

---

### Official Review · Reviewer_fprX · 2023-10-30

**Soundness:** 2 fair
**Presentation:** 2 fair
**Contribution:** 2 fair
**Rating:** 3
**Confidence:** 1

**Summary:**

The paper aims to quantify the inter-agent influences in MARL principally by proposing quantitative measurements on stop state values.

**Strengths:**

The paper is written with good clarity, with clear assumptions and definitions.

**Weaknesses:**

- As a reader, I'm still not convinced how useful is the proposed measurement. More theoretical or empirical evidence is needed in order to demonstrate their utilities.
- Some of the preliminaries needs to be added, such as the definition of Q values with differing actions.

**Questions:**

I view the proposed measurement as an alternative way of representing the reward, which changes the problem structure of MMDP. In order to argue for correctness, one would have to prove that the solution to the new problem is exactly the solution to the original problem. How would you relate the two problems?

---

> ### Author Response · Authors · 2023-11-20
> **Influence measurement functions and the underlying objective**
>
> Thank you for your review and the points you have raised. We appreciate the opportunity to clarify aspects of our work and to address your concerns regarding the utility and theoretical foundation of our proposed measurements.
>
> It is important to clarify that influence measurement functions in our work are descriptive inherent properties of a multi-agent Markov Decision Process. They are not intended to serve as objectives for the agents within the system. Therefore, our approach does not alter the fundamental problem. Rather, it provides an additional lens through which the inter-agent dynamics can be understood and quantified. The solutions derived from our measurements are congruent with the solutions to the original problem, offering insights into the influence dynamics without changing the core objective, such as finding a Nash equilibrium.
> While our focus is not on altering the objective function itself, we acknowledge that other works have successfully used such measurements to augment the objective function, for instance, in guiding exploration or enhancing cooperation among agents [1, 2]. This demonstrates the potential utility of influence measurements beyond their descriptive role.
> Our primary contribution lies in defining a general theoretical framework for describing influence measurement functions. We introduce TIM and SIM as instances within the framework, along with algorithms that guarantee convergence and provide error bounds for these measurements. Furthermore, we demonstrate the application of these theoretical results in complex practical settings, illustrating the utility and applicability of our approach.
> We hope these clarifications address your concerns.
>
> ### Bibliography:
>
> [1]: Tonghan Wang, Jianhao Wang, Yi Wu, Chongjie Zhang, Influence-based multi-agent exploration, ICLR2020
>
> [2]: Tonghan Wang, Liang Zeng, Weijun Dong, Qianlan Yang, Yang Yu, Chongjie Zhang, Context-Aware Sparse Deep Coordination Graphs, ICLR2022

---

> > ### Author Response · Authors · 2023-11-22
> > **End of discussion period**
> >
> > In light of the end of the author-reviewer discussion period, we would again highlight our response. We hope that we adequately addressed your concerns and convinced you to change your score accordingly.

---

### Official Review · Reviewer_dun1 · 2023-11-02

**Soundness:** 3 good
**Presentation:** 2 fair
**Contribution:** 2 fair
**Rating:** 3
**Confidence:** 3

**Summary:**

This paper studies the problem of quantifying the amount of influence an agent can have over other agents in multiagent reinforcement learning settings. Its main goal is to reliably detect the inherent influence structure of an environment given a specific policy. It’s main contributions include 1) a unified way to represent a multiagent system’s inherent influence structure, irrespective of the reward setting and overall objective; 2) introducing the total impact measurement and state impact measurement to quantify the overall and state-dependent influence structure respectively in multiagent average reward settings; 3) a decentralized algorithm with stability analysis and convergence guarantees together with empirical evaluation.

**Strengths:**

1.	Investigated an important problem --- quantifying agents’ influence among each other --- for MARL.
2.	Both empirical evaluation and theoretical analysis are provided to justify the effectiveness of the proposed metrics

**Weaknesses:**

1.	The novelty of this work was not fully justified.
There are many work study the structure influence of the individual agents for multiagent system. E.g,

Frans A. Oliehoek, Stefan Witwicki, Leslie P. Kaelbling, A Sufficient Statistic for Influence in Structured Multiagent Environments, JAIR2021

Thomas Spooner, Nelson Vadori, Sumitra Ganesh, Factored Policy Gradients: Leveraging Structure for Efficient Learning in MOMDPs, NeurIPS2021

If would be more convincing if more literature can be reviewed and related methods can be compared.

2.	Some technical details (e.g, stochastic iteration approximation) are introduced, but not clearly explained.

3.	The usefulness of the methods if not fully justified.
The paper is mainly investigating the proposed influence quantification metric. It is understandable that this is the first step towards deploying them for more applications related to MARL. It might be better if some comparison to some MARL methods (especially those have considered the influence structure like the one mentioned above) can be provided in some simple domains.

**Questions:**

1.	Section 3.2 provides some basic assumptions and theories related to stochastic approximations. It would be better if some intuitive explanations can be provided to help reader better understand background of the proposed metrics
2.	In section 6.1, “using the format of Zhang et al,…” what exactly it the format? It might be better to provide more details of the environment.
3.	Some notations are a little confusing, e.g, a^{-i}  and {a^{bar}}^{i}, they seem to refer to different things, but looks quite similar at first glance.
4.	Why emphasize averaged reward settings given that the proposed influence measurement can be applied to discounted reward settings as well?
5.	In section 6.2, “deviating from the original game, we employ one-sided penalty system…” Why need to deviate from the original game?
6.	What is L_add?

---

> ### Author Response · Authors · 2023-11-20
> **Broader Work on Influence in MARL**
>
> We highly appreciate the thorough review and constructive feedback. Your insights have been invaluable in refining our paper. Below, we address each of the concerns and questions raised.
> ### Broader Work on Influence in MARL
> The question of influence in MARL systems is indeed intricate and complex. Clearly, our influence measurement functions do not fully encapsulate the entire spectrum of influence. However, these functions represent a rich framework for unifying non-binary measurements that depend on the specific environment and the current joint policy.
> The work of Olihoek et al. [1] provides an intricate belief system about the interdependencies in multi-agent systems, comparable to the concept of Multi-Agent Influence Diagrams (MAIDs) introduced by Koller and Milch [3]. While their approach offers a richer framework and naturally provides ways to leverage its structure to search principled solution concepts like Nash equilibria, it faces challenges in tractability, even in small settings, due to the incorporation of strategic reasoning over the strategy space. In contrast, our influence measurement functions, being properties of the current joint policy, offer a descriptive snapshot of the system's state. This makes them more tractable than IBA or MAIDs. The trade-off, however, is that it leaves open questions regarding their utilization for enhancing learning. We've noted previous work [4, 5, 6] that has started to explore the applications of these functions, and we believe there are further opportunities in this area for future research.
> The work by Spooner et al. [2] primarily follows the line of binary influence representations, as seen in coordination graphs [7]. This approach simplifies the analysis but imposes a strong assumption about the environment. As we discuss in Section 4, complete independence among agents is rare. Our methodology advocates for a more nuanced view. For instance, in the coin game scenario, despite minimal influence by some agents, the “influence matrix” by Spooner et al. would consist entirely of ones, indicating each agent's ability to affect others' rewards. This lack of nuance means that approaches like Spooner et al.'s do not offer additional insights into the influence structure of such games. Their optimization method would default to optimizing a single controller, overlooking the subtleties of agent interactions that our approach captures.
> In conclusion, while our influence measurement functions may not encompass all aspects of influence in MARL, they provide a significant step toward understanding and quantifying inter-agent dynamics. This paves the way for future research to build upon and utilize these measures in more complex and varied MARL environments. We edited our comments about future work to point this out more clearly.
>
> ### Bibliography:
>
> [1]: Frans A. Oliehoek, Stefan Witwicki, Leslie P. Kaelbling, A Sufficient Statistic for Influence in Structured Multiagent Environments, JAIR2021
>
> [2]: Thomas Spooner, Nelson Vadori, Sumitra Ganesh, Factored Policy Gradients: Leveraging Structure for Efficient Learning in MOMDPs, NeurIPS2021
>
> [3]: Daphne Koller, and Brian Milch, Multi-agent influence diagrams for representing and solving games, Games and Economic Behavior, volume 45, 2003
>
> [4]: Tonghan Wang, Jianhao Wang, Yi Wu, Chongjie Zhang, Influence-based multi-agent exploration, ICLR2020
>
> [5]: Jaques et al., Social Influence as Intrinsic Motivation for Multi-Agent Deep Reinforcement Learning, ICML2019
>
> [6]: Tonghan Wang, Liang Zeng, Weijun Dong, Qianlan Yang, Yang Yu, Chongjie Zhang, Context-Aware Sparse Deep Coordination Graphs, ICLR2022
>
> [7]: Carlos Guestrin, Shobha Venkataraman, and Daphne Koller. Context-Specific Multiagent Coordination and Planning with Factored MDPs, ICML2002

---

> ### Author Response · Authors · 2023-11-20
> **Addressing raised Questions**
>
> ### Section 3.2: Stochastic Approximations and Intuitive Explanations
> Although there is no proof in the main text, we estimate the background information provided in Section 3.2 to be essential. It helps the reader understand the rationale and design principles behind our proposed measurement algorithms.
> Section 6.1: Environment Details and Reference to Zhang et al.
> We acknowledge that our reference to Zhang et al. was misleading, implying a need to consult their paper for details. That is not the case, as we provide a detailed description of the environment in the appendix. We clarified the reference to this addition in the main text, resolving ambiguity.
> ### Notation Clarification
> Agreed, some of our notations were confusing. We have revised these notations for clarity and consistency.
> ### Emphasis on Average Reward Settings
> Our decision to emphasize average reward settings stems from a noticeable gap in existing research. To our knowledge, there are currently no interdependency measures that align with our definition of an influence measurement function specifically for average reward settings. This contrasts with the discounted reward setting, where measures like the Value of Interaction [4] already exist. Our work aims to fill this gap by introducing new metrics tailored to the average reward setting, along with corresponding theoretical guarantees.
> Additionally, while the underlying definition of influence measurement functions is independent of the setting, TIM and SIM are not. They are specifically designed for the average reward setting. Although there are natural extensions of these measures to the discounted reward setting, the theoretical foundations and proofs of TIM and SIM are intrinsically linked to properties unique to the average reward setting. This setting-specific focus allows for a more rigorous and targeted analysis, ensuring that the theoretical guarantees we provide are robust and applicable within this context.
> ### Section 6.2: Deviation from the Original Game
> In our study, we chose to deviate from the original symmetric game format by implementing a one-sided penalty system in the coin game. The primary reason for this modification was to expose and highlight the differences in interdependencies among the agents. In a symmetric game, one might expect the influence patterns to also be symmetric. However, our aim was to explore and demonstrate more complex influence dynamics. By introducing asymmetry through one-sided penalties, we were able to provide a clearer semantic explanation of these complex dynamics. This decision and its rationale have been more explicitly articulated in the revised paper for greater clarity.
> ### Explanation of L_add
> In our study, L_add is used to control the number of additional influences in our randomly generated environment. This parameter allows us to precisely adjust how much influence each agent has over others. By default, we assume that each agent can influence its own expected return. The value of L_add quantifies the extent of additional influences that go beyond this basic assumption. For example, L_add=0 indicates a scenario where agents do not exert any influence on each other. Conversely, L_add=20 in a system of five agents signifies a situation where every agent has the capacity to influence every other agent.
> We hope that these clarifications and modifications address the concerns raised and demonstrate the novelty and applicability of our work in the field of multiagent reinforcement learning. We are grateful for the opportunity to enhance our paper based on your insightful feedback.
>
> ### Bibliography:
>
> [4]: Tonghan Wang, Jianhao Wang, Yi Wu, Chongjie Zhang, Influence-based multi-agent exploration, ICLR2020

---

> > ### Author Response · Authors · 2023-11-22
> > **End of discussion period**
> >
> > In light of the end of the author-reviewer discussion period, we would again highlight our response. We hope that we adequately addressed your concerns and convinced you to change your score accordingly.

---

### Official Review · Reviewer_gDUL · 2023-11-09

**Soundness:** 3 good
**Presentation:** 2 fair
**Contribution:** 3 good
**Rating:** 6
**Confidence:** 4

**Summary:**

The paper mainly discusses the problem of measuring the influence levels between agents interacting in an environment in the average reward setting. Authors introduce two performance metrics: the total impact measurement and state impact measurement. The paper proposes measuring the effect of other agents by dividing the set of agents into subsystems, and measuring the effect of agents in this subsystem.

**Strengths:**

The paper is written clearly and the two measures are well-defined. The empirical results also support the theoretical results presented in section 5.

**Weaknesses:**

It seems like definition 4.1 is related to the potential function in a Markov potential game, and I would ask the authors if they can discuss any connections between the matrix entries for one state-action and a particular potential function. The formulation of the problem itself as a multi-agent MDP is indeed a Markov game.

**Questions:**

How does the impact scale differ from a potential function defined as the deviation in policies between agent i and the other agents -i in the subset of agents?

---

> ### Author Response · Authors · 2023-11-20
> **Connections to Potential Functions**
>
> Thank you for your valuable feedback and insightful observations regarding our paper. We appreciate the opportunity to address your questions and clarify certain aspects of our work.
> ### Connections to Potential Functions
> Your observation regarding the connection between our proposed measurement functions and potentials is fitting. We agree that there is a conceptual relationship, but also key differences that distinguish our approach.
> Our state- or total-influence measurement functions identify the existence of a potential in agent i’s state-action function due to agent j’s behavior. Specifically, if agent j can influence agent i, this influence manifests as a non-zero potential in our framework.
> The Markov potential function, in contrast, is an agent-independent mapping that captures state-value potentials for unilateral deviations in policy by any agent. This means that if agent i changes its behavior, the change in its state-value is indicated by the potential function. However, this function does not account for the impact of agent j’s policy changes on agent i’s state-value. Therefore, it does not adequately represent the interdependencies between agents.
> The Markov potential function encapsulates the agents’ incentives, and analyzing the local optima of this function can lead to finding Nash Equilibria. This is a strong property and not universally applicable. Our influence measurement functions describe potentials in strategic deviations among all agents, offering a broader perspective without limiting the strategic complexity inherent in the Markov game. Therefore, the Total Impact Measurement (TIM) and State Impact Measurement (SIM) proposed in Section 5 exist in every Markov game within the average reward setting, irrespective of the existence of a Markov potential function. Furthermore, even in cases where a Markov potential function exists, it may not sufficiently express the complex interdependencies among agents.
>
> ### Clarification on Terminology: Multi-agent MDP and Markov Game
> We agree that the common term is Markov or Stochastic game. We have added a comment in the revised manuscript to highlight this connection. We decided to use the terminology provided by Zhang et al. [1] for two reasons. First, we wanted to emphasize the underlying MDP properties as our results and concepts rely heavily on those. Second, it decreases ambiguity regarding the specifics of the reward setting so that readers do not assume that the theoretical results hold for other settings that usually fall under the Markov game term.
> We hope that these clarifications and modifications address your concerns. Thank you once again for your constructive feedback.
>
> Bibliography:
> [1]: Zhang et al., Fully decentralized Multi-Agent Reinforcement Learning with Networked Agents, ICML2018

---

> > ### Author Response · Authors · 2023-11-22
> > **End of discussion period**
> >
> > In light of the end of the author-reviewer discussion period, we would again highlight our response. We hope that we adequately addressed your concerns and convinced you to change your score accordingly.

---

### Meta-Review · Area_Chair_Wwf7 · 2023-12-08

**Metareview:**

a) claims: The paper defines two measures of the influence that one agent can have on another's expected return in a MARL setting, and gives algorithms for approximating these quantities.

b) strengths: Most reviewers agreed that the question of estimating between-agent influence is an important one.

c) weaknesses: Two reviewers had serious concerns about a lack of engagement with the literature. One suggested a number of other papers that attempt similar estimation tasks that should be discussed as related work.  Another noted that the empirical evaluations did not compare to any baselines.  Multiple reviewers were unsure what the proposed measures would be useful for.

**Justification For Why Not Higher Score:**

The main concerns behind my rating were the issues of comparisons to prior work, and the need for a clearer articulation of what should be done with the derived measurements.

**Justification For Why Not Lower Score:**

n/a

---

### Decision · Program_Chairs · 2024-01-16

Reject